# Robust model-based analysis of single-particle tracking experiments with Spot-On

Anders S Hansen[1,2†]*, Maxime Woringer[1,3,4†], Jonathan B Grimm[5], Luke D Lavis[5], Robert Tjian[1,2]*, Xavier Darzacq[1]*

[1]Department of Molecular and Cell Biology, Li Ka Shing Center for Biomedical and Health Sciences, CIRM Center of Excellence, University of California, Berkeley, Berkeley, United States; [2]Howard Hughes Medical Institute, Berkeley, United States; [3]Unité Imagerie et Modélisation, Institut Pasteur, Paris, France; [4]UPMC Univ Paris 06, Sorbonne Universités, Paris, France; [5]Janelia Research Campus, Howard Hughes Medical Institute, Ashburn, United States

**Abstract** Single-particle tracking (SPT) has become an important method to bridge biochemistry and cell biology since it allows direct observation of protein binding and diffusion dynamics in live cells. However, accurately inferring information from SPT studies is challenging due to biases in both data analysis and experimental design. To address analysis bias, we introduce 'Spot-On', an intuitive web-interface. Spot-On implements a kinetic modeling framework that accounts for known biases, including molecules moving out-of-focus, and robustly infers diffusion constants and subpopulations from pooled single-molecule trajectories. To minimize inherent experimental biases, we implement and validate stroboscopic photo-activation SPT (spaSPT), which minimizes motion-blur bias and tracking errors. We validate Spot-On using experimentally realistic simulations and show that Spot-On outperforms other methods. We then apply Spot-On to spaSPT data from live mammalian cells spanning a wide range of nuclear dynamics and demonstrate that Spot-On consistently and robustly infers subpopulation fractions and diffusion constants.

DOI: https://doi.org/10.7554/eLife.33125.001

*For correspondence:
anders.sejr.hansen@berkeley.edu (ASH);
jmlim@berkeley.edu (RT);
darzacq@berkeley.edu (XD)

†These authors contributed equally to this work

## Introduction

Advances in imaging technologies, genetically encoded tags and fluorophore development have made single-particle tracking (SPT) an increasingly popular method for analyzing protein dynamics (*Liu et al., 2015*). Recent biological applications of SPT have revealed that transcription factors (TFs) bind mitotic chromosomes (*Teves et al., 2016*), how Polycomb interacts with chromatin (*Zhen et al., 2016*), that 'pioneer factor' TFs bind chromatin dynamically (*Swinstead et al., 2016*), that TF binding time correlates with transcriptional activity (*Loffreda et al., 2017*) and that different nuclear proteins adopt distinct target search mechanisms (*Izeddin et al., 2014*; *Rhodes et al., 2017*). Compared with indirect and bulk techniques such as Fluorescence Recovery After Photobleaching (FRAP) or Fluorescence Correlation Spectroscopy (FCS), SPT is often seen as less biased and less model-dependent (*Goulian and Simon, 2000*; *Mueller et al., 2013*; *Shen et al., 2017*). In particular, SPT makes it possible to directly follow single molecules over time in live cells and has provided clear evidence that proteins often exist in several subpopulations that can be characterized by their distinct diffusion coefficients (*Mueller et al., 2013*; *Shen et al., 2017*). For example, nuclear proteins such as TFs and chromatin binding proteins typically show a quasi-immobile chromatin-bound fraction and a freely diffusing fraction inside the nucleus. However, while SPT of slow-diffusing membrane proteins is an established technology (*Weimann et al., 2013*), 2D-SPT of proteins

**eLife digest** Proteins, the molecules that make up the cells' internal machinery, are responsible for almost every process that keeps cells alive. Watching how proteins move and interact within a living cell can help scientists to better understand these biological mechanisms. Single-particle tracking is a recent technique that makes these observations possible by taking 'live' recordings of individual proteins in a cell. Typically, the goal of a single-particle tracking experiment is to assign proteins into groups, or subpopulations, based on the way they move in the cell. For example, one subpopulation may be bound to other cellular structures, a second moving freely at a high speed, and a third diffusing slowly. This informs on the biological roles of the proteins.

The method involves an experimental stage and an analysis stage. During the experiment, proteins of interest are labeled with a small dye molecule that produces light when excited by a laser. The laser then illuminates the cell, stimulating all the labels in a thin layer. The position of each molecule is then determined with a microscope and a 'snapshot' taken. By repeating this process over multiple images, the movement of each molecule over time can be tracked. However, experimental problems can make the interpretation difficult. Motion blurring takes place when the proteins move so fast they appear as blurs in the images; tracking errors happen when so many proteins are present in the same space their trajectories overlap.

Here, Hansen, Woringer et al. combine two pre-existing methods to improve the experimental set-up. Using lasers that flash like a strobe light reduces motion blurring by essentially taking snapshots of the proteins at short time intervals. Tracking errors are addressed by a technique whereby only one protein at a time produces light.

Once the images are obtained and analyzed to yield trajectories, the trajectories themselves need to be analyzed to determine the number and properties of the protein subpopulations. Several factors can skew this analysis stage. For example, there is often a bias against fast-moving particles because the laser only lights up a thin layer of the cell. The proteins travelling slowly stay in focus long enough to be detected across many images; the fast ones quickly move out of the layer and are therefore counted less often. Hansen, Woringer et al. designed a free and user-friendly algorithm package called Spot-On to correct for this issue. Spot-On was thoroughly benchmarked against other solutions, demonstrating both its accuracy and robustness.

Single-particle tracking can lead to misleading results if used incorrectly. It is essential to publically share solutions that help make this technique more rigorous, especially since a growing number of scientists have already started to use the method.

DOI: https://doi.org/10.7554/eLife.33125.002

freely diffusing inside a 3D nucleus introduces several biases that must be corrected for in order to obtain accurate estimates of subpopulations. First, while a frame is acquired, fast-diffusing molecules move and spread out their emitted photons over multiple pixels causing a '*motion-blur*' artifact (*Berglund, 2010*; *Deschout et al., 2012*; *Frost et al., 2012*; *Goulian and Simon, 2000*; *Izeddin et al., 2014*), whereas immobile or slow-diffusing molecules resemble point spread functions (PSFs; *Figure 1A*). This results in under-counting of the fast-diffusing subpopulation. Second, high particle densities tend to cause *tracking errors* when localized molecules are connected into trajectories. This can result in incorrect displacement estimates (*Figure 1B*). Third, since SPT generally employs 2D imaging of 3D motion, immobile or slow-diffusing molecules will generally remain in-focus until they photobleach and therefore exhibit long trajectories, whereas fast-diffusing molecules in 3D rapidly move out-of-focus, thus resulting in short trajectories (we refer to this as '*defocalization*'; *Figure 1C*). This results in a time-dependent under-counting of fast-diffusing molecules (*Goulian and Simon, 2000*; *Kues and Kubitscheck, 2002*). Fourth, SPT *analysis methods* themselves may introduce biases; to avoid this, an accurate and validated method is needed (*Figure 1D*).

Here, we introduce an integrated approach to overcome all four biases. The first two biases must be minimized at the data acquisition stage and we describe an experimental SPT method to do so (spaSPT), whereas the latter two can be overcome using a previously developed kinetic modeling framework (*Hansen et al., 2017*; *Mazza et al., 2012*) now extended and implemented in Spot-On.

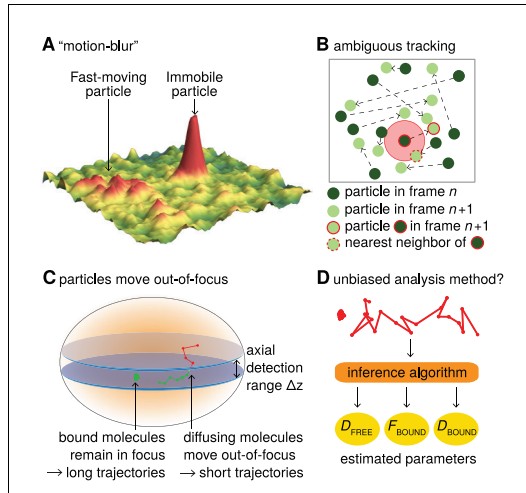

**Figure 1.** Bias in single-particle tracking (SPT) experiments and analysis methods. (**A**) 'Motion-blur' bias. Constant excitation during acquisition of a frame will cause a fast-moving particle to spread out its emission photons over many pixels and thus appear as a motion-blur, which make detection much less likely with common PSF-fitting algorithms. In contrast, a slow-moving or immobile particle will appear as a well-shaped PSF and thus readily be detected. (**B**) Tracking ambiguities. Tracking at high particle densities prevents unambiguous connection of particles between frames and tracking errors will cause displacements to be misidentified. (**C**) Defocalization bias. During 2D-SPT, fast-moving particles will rapidly move out-of-focus resulting in short trajectories, whereas immobile particles will remain in-focus until they photobleach and thus exhibit very long trajectories. This results in a bias toward slow-moving particles, which must be corrected for. (**D**) Analysis method. Any analysis method should ideally avoid introducing biases and accurately correct for known biases in the estimation of subpopulation parameters such as $D_{FREE}$, $F_{BOUND}$, $D_{BOUND}$.
DOI: https://doi.org/10.7554/eLife.33125.003

Spot-On is available as a web-interface (https://SpotOn.berkeley.edu) as well as Python and Matlab packages.

## Results

### Overview of Spot-On

Spot-On is a user-friendly web-interface that pedagogically guides the user through a series of quality-checks of uploaded datasets consisting of pooled single-molecule trajectories. It then performs kinetic model-based analysis that leverages the histogram of molecular displacements over time to infer the fraction and diffusion constant of each subpopulation (*Figure 2*). Spot-On does not directly analyze raw microscopy images, since a large number of localization and tracking algorithms exist that convert microscopy images into single-molecule trajectories (for a comparison of particle tracking methods, see (*Chenouard et al., 2014*); moreover, Spot-On can be one-click interfaced with TrackMate (*Tinevez et al., 2017*), which allows inspection of trajectories before uploading to Spot-On).

To use Spot-On, a user uploads their SPT trajectory data in one of several formats (*Figure 2*). Spot-On then generates useful meta-data for assessing the quality of the experiment (e.g. localization density, number of trajectories etc.). Spot-On also allows a user to upload multiple datasets (e.g. different replicates) and merge them. Spot-On then calculates and displays histograms of displacements over multiple time delays. The next step is model fitting. Spot-On models the distribution of displacements for each subpopulation using Brownian motion under steady-state conditions without state transitions (full model description in Materials and Methods). Spot-On also accounts for localization errors (either user-defined or inferred from the SPT

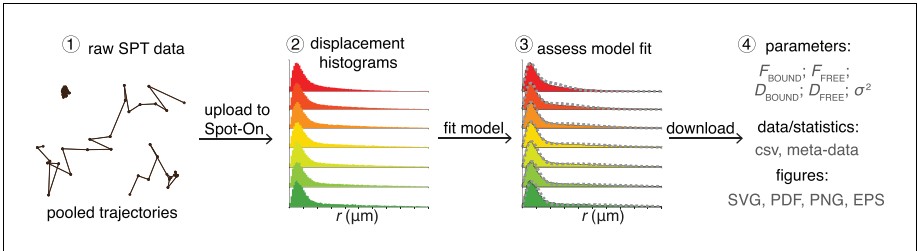

**Figure 2.** Overview of Spot-On interface. To use Spot-On, a user uploads raw SPT data in the form of pooled SPT trajectories to the Spot-On web-interface. Spot-On then calculates displacement histograms. The user inputs relevant experimental descriptors and chooses a model to fit. After model-fitting, the user can then download model-inferred parameters, meta-data and download publication-quality figures.
DOI: https://doi.org/10.7554/eLife.33125.004

data). Crucially, Spot-On corrects for defocalization bias (*Figure 1C*) by explicitly calculating the probability that molecules move out-of-focus as a function of time and their diffusion constant (*Video 1*). In fact, Spot-On uses the gradual loss of freely diffusing molecules over time as additional information to infer the diffusion constant and size of each subpopulation.

Spot-On considers either 2 or 3 subpopulations. For instance, TFs in nuclei can generally exist in both a chromatin-bound state characterized by slow diffusion and a freely diffusing state associated with rapid diffusion. In this case, a 2-state model is generally appropriate ('bound' vs. 'free'). Spot-On allows a user to choose their desired model and parameter ranges and then fits the model to the data. Using the previous example of TF dynamics, this allows the user to infer the bound fraction and the diffusion constants. Finally, once a user has finished fitting an appropriate model to their data, Spot-On allows easy download of publication-quality figures and relevant data (*Figure 2*; Full tutorial in *Supplementary file 1*).

## Validation of Spot-On using simulated SPT data and comparison to other methods

We first evaluated whether Spot-On could accurately infer subpopulations (*Figure 1D*) and successfully account for known biases (*Figure 1C*) using simulated data. We compared Spot-On to a popular alternative approach of first fitting the mean square displacement (MSD) of individual trajectories of a minimum length and then fitting the distribution of estimated diffusion constants (we refer to this as 'MSD$_i$') as well as a sophisticated Hidden-Markov Model-based Bayesian inference method (vbSPT) (*Persson et al., 2013*). Since most SPT data is collected using highly inclined illumination (*Tokunaga et al., 2008*) (HiLo), we simulated TF binding and diffusion dynamics (2-state model: 'bound vs. free') confined inside a 4 μm radius mammalian nucleus under realistic HiLo SPT experimental settings subject to a 25 nm localization error (*Figure 3—figure supplement 1*). We considered the effect of the exposure time (1 ms, 4 ms, 7 ms, 13 ms, 20 ms), the free diffusion constant (from 0.5 μm$^2$/s to 14.5 μm$^2$/s in 0.5 μm$^2$/s increments) and the bound fraction (from 0% to 95% in 5% increments) yielding a total of 3480 different conditions that span the full range of biologically plausible dynamics (*Figure 3—figure supplements 2–3*; Appendix 1).

Spot-On accurately inferred subpopulation sizes with minimal error (*Figure 3A–B*, *Table 1*), but slightly underestimated the diffusion constant (−4.8%; *Figure 3B*; *Table 1*). However, this underestimate was due to particle confinement inside the nucleus: Spot-On correctly inferred the diffusion constant when the confinement was relaxed (*Figure 3—figure supplement 4*; 20 μm nuclear radius instead of 4 μm). This emphasizes that diffusion constants measured by SPT inside cells should be viewed as apparent diffusion constants. In contrast, the MSD$_i$ method failed under most conditions regardless of whether all trajectories were used (MSD$_i$ (all)) or a fitting filter applied (MSD$_i$ ($R^2$ >0.8); *Figure 3A–B*; *Table 1*). vbSPT performed almost as well as Spot-On for slow-diffusing proteins, but showed larger deviations for fast-diffusing proteins (*Figure 3—figure supplements 2–3*).

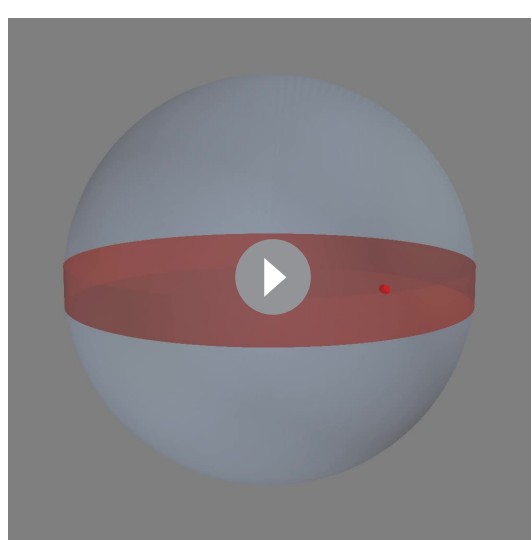

**Video 1.** Related to *Figure 1*. Illustration of defocalization bias. Illustration of a single-particle tracking experiment with two subpopulations (one 'immobile', D = 0.001 μm$^2$/s, the other 'free', D = 4 μm$^2$/s with a 1:1 ratio, observed using 20 ms time interval). The red region corresponds to the axial detection range (1 μm) and molecules randomly appear when they photo-activate. For each trajectory, the detected localizations inside the detection range are shown as red spheres and undetected localizations outside the detection range are shown as white spheres. Each particle has a mean lifetime of 15 frames, 25 nm localization error and trajectories consisting of at least two frames are plotted. Epi illumination is assumed. The SPT data was simulated and plotted using simSPT (available at https://gitlab.com/tjian-darzacq-lab/simSPT).
DOI: https://doi.org/10.7554/eLife.33125.005

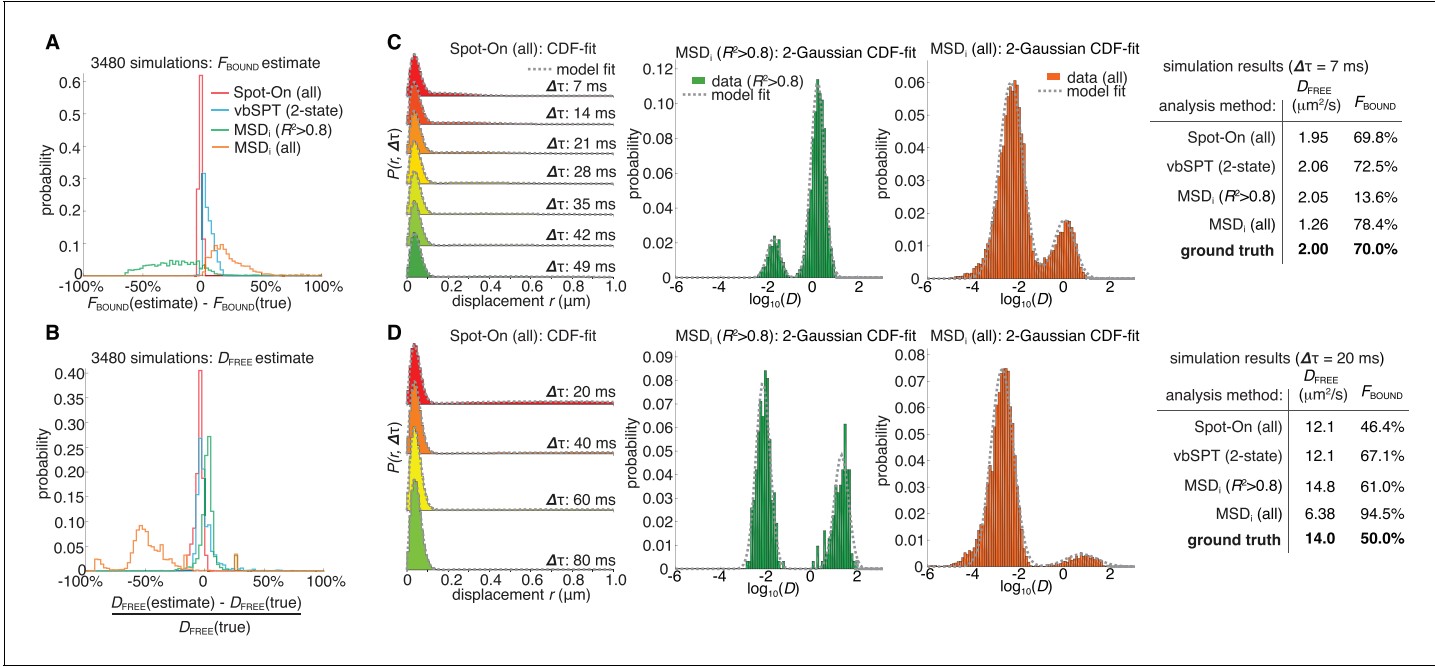

**Figure 3.** Validation of Spot-On using simulations and comparisons to other methods. (A–B) Simulation results. Experimentally realistic SPT data was simulated inside a spherical mammalian nucleus with a radius of 4 μm subject to highly-inclined and laminated optical sheet illumination (*Tokunaga et al., 2008*) (HiLo) of thickness 4 μm illuminating the center of the nucleus. The axial detection window was 700 nm with Gaussian edges and particles were subject to a 25 nm localization error in all three dimensions. Photobleaching corresponded to a mean trajectory length of 4 frames inside the HiLo sheet and 40 outside. 3480 experiments were simulated with parameters of $D_{FREE}$=[0.5;14.5] in steps of 0.5 μm$^2$/s and $F_{BOUND}$=[0;95% in steps of 5% and the frame rate correspond to $\Delta\tau$=[1,4,7,10,13,20] ms. Each experiment was then fitted using Spot-On, using vbSPT (maximum of 2 states allowed) (*Persson et al., 2013*), MSD$_i$ using all trajectories of at least five frames (MSD$_i$ (all)) or MSD$_i$ using all trajectories of at least five frames where the MSD-curvefit showed at least $R^2$ >0.8 (MSD$_i$ ($R^2$ >0.8)). (A) shows the distribution of absolute errors in the $F_{BOUND}$–estimate and (B) shows the distribution of relative errors in the $D_{FREE}$–estimate. (C) Single simulation example with $D_{FREE}$ = 2.0 μm$^2$/s; $F_{BOUND}$ = 70%; 7 ms per frame. The table on the right uses numbers from CDF-fitting, but for simplicity the fits to the histograms (PDF) are shown in the three plots. (D) Single simulation example with $D_{FREE}$ = 14.0 μm$^2$/s; $F_{BOUND}$ = 50%; 20 ms per frame. Full details on how SPT data was simulated and analyzed with the different methods is given in Appendix 1.

DOI: https://doi.org/10.7554/eLife.33125.006

The following figure supplements are available for figure 3:

**Figure supplement 1.** Overview of SPT simulations.
DOI: https://doi.org/10.7554/eLife.33125.007

**Figure supplement 2.** Comparison of Spot-On, vbSPT and MSDi estimates of $D_{FREE}$ and $F_{BOUND}$ to ground-truth simulation results inside a 4 μm radius nucleus.
DOI: https://doi.org/10.7554/eLife.33125.008

**Figure supplement 3.** Representative fits for Spot-On, vbSPT and MSDi to ground-truth simulations.
DOI: https://doi.org/10.7554/eLife.33125.009

**Figure supplement 4.** Comparison of Spot-On, vbSPT and MSDi estimates of $D_{FREE}$ and $F_{BOUND}$ to ground-truth simulations inside a 20 μm radius nucleus.
DOI: https://doi.org/10.7554/eLife.33125.010

**Figure supplement 5.** Effect of defocalization bias correction.
DOI: https://doi.org/10.7554/eLife.33125.011

**Figure supplement 6.** Evaluation of the 3-states model.
DOI: https://doi.org/10.7554/eLife.33125.012

**Figure supplement 7.** Sensitivity of Spot-On to the axial detection range estimate.
DOI: https://doi.org/10.7554/eLife.33125.013

**Figure supplement 8.** Sensitivity of Spot-On to the number of time points considered.
DOI: https://doi.org/10.7554/eLife.33125.014

**Figure supplement 9.** Comparison of Spot-On and MSDi estimates of $D_{FREE}$ and $F_{BOUND}$ to ground-truth simulation results inside a 4 μm radius nucleus using PDF-fitting.
DOI: https://doi.org/10.7554/eLife.33125.015

*Figure 3 continued on next page*

*Figure 3 continued*

**Figure supplement 10.** Sensitivity of Spot-On to state changes and comparison with vbSPT.
DOI: https://doi.org/10.7554/eLife.33125.016
**Figure supplement 11.** Robustness of localization error estimates from Spot-On.
DOI: https://doi.org/10.7554/eLife.33125.017
**Figure supplement 12.** Sensitivity of Spot-On, vbSPT and MSD$_i$ ($R^2$ >0.8) to sample size.
DOI: https://doi.org/10.7554/eLife.33125.018

To illustrate how the methods could give such divergent results when run on the same SPT data, we considered two example simulations (*Figure 3C–D*; more examples in *Figure 3—figure supplement 3*). First, we considered a mostly bound and relatively slow diffusion case ($D_{FREE}$: 2.0 μm$^2$/s; $F_{BOUND}$: 70%; Δτ: 7 ms; *Figure 3C*). Spot-On and vbSPT accurately inferred both $D_{FREE}$ and $F_{BOUND}$. In contrast, MSD$_i$ ($R^2$ > 0.8) greatly underestimated $F_{BOUND}$ (13.6% vs. 70%), whereas MSD$_i$ (all) slightly overestimated $F_{BOUND}$. Since MSD$_i$-based methods apply two thresholds (first, minimum trajectory length: here five frames; second, filtering based on $R^2$) in many cases less than 5% of all trajectories passed these thresholds and this example illustrate how sensitive MSD$_i$-based methods are to these thresholds. Note that although we show the fits to the probability density function since this is more intuitive (PDF; histogram), we performed the fitting to the cumulative distribution function (CDF). Second, we considered an example with a slow frame rate and fast diffusion, such that the free population rapidly moves out-of-focus ($D_{FREE}$: 14.0 μm$^2$/s; $F_{BOUND}$: 50%; Δτ: 20 ms; *Figure 3D*). Spot-On again accurately inferred $F_{BOUND}$, and slightly underestimated $D_{FREE}$ due to high nuclear confinement (*Figure 3—figure supplement 4*). Although vbSPT generally performed well, because it does not correct for defocalization bias (vbSPT was developed for bacteria, where defocalization bias is minimal), vbSPT strongly overestimated $F_{BOUND}$ in this case (*Figure 3D*). Consistent with this, Spot-On without defocalization-bias correction also strongly overestimates the bound fraction (*Figure 3—figure supplement 5*). We conclude that correcting for defocalization bias is critical. The MSD$_i$-based methods again gave divergent results despite seemingly fitting the data well. Thus, a good fit to a histogram of log(D) does not necessarily imply that the inferred $D_{FREE}$ and $F_{BOUND}$ are accurate. A full discussion and comparison of the methods is given in Appendix 1. Finally, we extended this analysis of simulated SPT data to three states (one 'bound', two 'free' states) and compared Spot-On and vbSPT. Spot-On again accurately inferred both the diffusion constants and subpopulation fractions of each population and slightly outperformed vbSPT (*Figure 3—figure supplement 6*).

Having established that Spot-On is accurate, we next tested whether it was also robust. Spot-On's ability to infer $D_{FREE}$ and $F_{BOUND}$ was robust to misestimates of the axial detection range of ~100–200 nm (*Figure 3—figure supplement 7*), was minimally affected by the number of time-points considered and fitting parameters (*Figure 3—figure supplements 8–9*; see also Appendix 2 for parameter considerations) and was not strongly affected by state changes (e.g. binding or unbinding) provided the time-scale of state changes is significantly longer than the frame rate (*Figure 3—figure supplement 10*). Moreover, Spot-On inferred the localization error with nanometer precision provided that a significant bound fraction is present (*Figure 3—figure supplement 11*). Finally, we sub-sampled the data sets and found that just ~3000 short trajectories (mean length ~3–4 frames) were sufficient for Spot-On to reliably infer the underlying dynamics (*Figure 3—figure supplement 12*). We conclude that Spot-On is robust.

Taken together, this analysis of simulated SPT data suggests that Spot-On successfully overcomes defocalization and analysis method biases (*Figure 1C–D*), accurately and robustly estimates subpopulations and diffusion constants across a wide range of dynamics and, finally, outperforms other methods.

## spaSPT minimizes biases in experimental SPT acquisitions

Having validated Spot-On on simulated data, which is not subject to experimental biases (*Figure 1A–B*), we next sought to evaluate Spot-On on experimental data. To generate SPT data with minimal acquisition bias we performed stroboscopic photo-activation SPT (spaSPT; *Figure 4A*), which integrates previously and separately published ideas to minimize experimental biases. First, spaSPT minimizes motion-blurring, which is caused by particle movement during the camera

**Table 1.** Summary of simulation results and comparison of methods.

The table shows the bias (mean error), 'std' (standard deviation) and 'iqr' (inter-quartile range: difference between the 75th and 25th percentile) for each method for all 3480 simulations. The left column shows the relative bias/std/iqr for the $D_{FREE}$-estimate and the right column shows the absolute bias/std/iqr for the $F_{BOUND}$-estimate.

| Analysis method | $D_{FREE}$ | | | $F_{BOUND}$ | | |
|---|---|---|---|---|---|---|
| | bias | std | iqr | bias | std | iqr |
| Spot-On (all) | −4.8% | 3.3% | 3.5% | −1.7% | 1.2% | 1.8% |
| vbSPT (2-state) | 0.8% | 12.5% | 6.8% | 5.0% | 4.6% | 6.1% |
| $MSD_i$ ($R^2 > 0.8$) | 8.0% | 28.5% | 4.9% | −20.6% | 26.4% | 32.1% |
| $MSD_i$ (all) | −39.6% | 41.8% | 19.0% | 22.0% | 15.8% | 17.8% |

DOI: https://doi.org/10.7554/eLife.33125.019

exposure time (*Figure 1A*), by using stroboscopic excitation (*Elf et al., 2007*; *Frost et al., 2012*). We found that the bright and photo-stable dyes PA-JF$_{549}$ and PA-JF$_{646}$ (*Grimm et al., 2016a*) in combination with the HaloTag ('Halo') labeling strategy made it possible to achieve a signal-to-background ratio greater than 5 with just 1 ms excitation pulses, thus providing a good compromise between minimal motion-blurring and high signal (*Figure 4B*). Second, spaSPT minimizes tracking errors (*Figure 1B*) by using photo-activation (*Figure 4A*) (*Grimm et al., 2016a*; *Manley et al., 2008*). Tracking errors are generally caused by high particles densities. Photo-activation allows tracking at extremely low densities (≤1 molecule per nucleus per frame) and thereby minimizes tracking errors (*Izeddin et al., 2014*), whilst at the same time generating thousands of trajectories. To consider the full spectrum of nuclear protein dynamics, we studied histone H2B-Halo (overwhelmingly bound; fast diffusion; *Figure 4C*), Halo-CTCF (*Hansen et al., 2017*) (largely bound; slow diffusion; *Figure 4D*) and Halo-NLS (overwhelmingly free; very fast diffusion; *Figure 4F*) in human U2OS cells and Halo-Sox2 (*Teves et al., 2016*) (largely free; intermediate diffusion; *Figure 4E*) in mouse embryonic stem cells (mESCs). We labeled Halo-tagged proteins in live cells with the HaloTag ligands PA-JF$_{549}$ or PA-JF$_{646}$ (*Grimm et al., 2016a*) and performed spaSPT using HiLo illumination (*Video 2*). To generate a large dataset to comprehensively test Spot-On, we performed 1064 spaSPT experiments across 60 different conditions.

## Validation of Spot-On using spaSPT data at different frame rates

First, we studied whether Spot-On could consistently infer subpopulations over a wide range of frame rates. We experimentally determined the axial detection range to be ~700 nm (*Figure 4—figure supplement 1*) and performed spaSPT at 200 Hz, 167 Hz, 134 Hz, 100 Hz, 74 Hz and 50 Hz using the four cell lines. Spot-On consistently inferred the diffusion constant (*Figure 4G*) and total bound fraction across the wide range of frame rates (*Figure 4H*). This is notable since all four proteins exhibit apparent anomalous diffusion (*Figure 4—figure supplement 2*) and this demonstrates that Spot-On is also robust to anomalous diffusion despite modeling Brownian motion. While the ground-truth is unknown when considering experiments, Spot-On gave biologically reasonable results: histone H2B was overwhelmingly bound and free Halo-3xNLS was overwhelmingly unbound (comparison with vbSPT: *Figure 4—figure supplement 3*). These results provide additional validation for the bias corrections implemented in Spot-On. We also note that although Spot-On was validated on spaSPT data, SPT data with non-photoactivatable dyes is also suitable for Spot-On analysis provided that the density is sufficiently low to minimize tracking errors (see also Appendix 3: "Which datasets are appropriate for Spot-On?"). Finally, we demonstrated above that just ~3000 short trajectories (mean length ~3–4 frames) were sufficient for Spot-On to accurately infer $D_{FREE}$ and $F_{BOUND}$ (*Figure 3—figure supplement 12*). Here we obtain well above 3000 trajectories per cell even at ~1 localization/frame. More generally, with spaSPT this should be generally achievable for all but the most lowly expressed nuclear proteins. Thus, this now makes it possible to study biological cell-to-cell variability in TF dynamics.

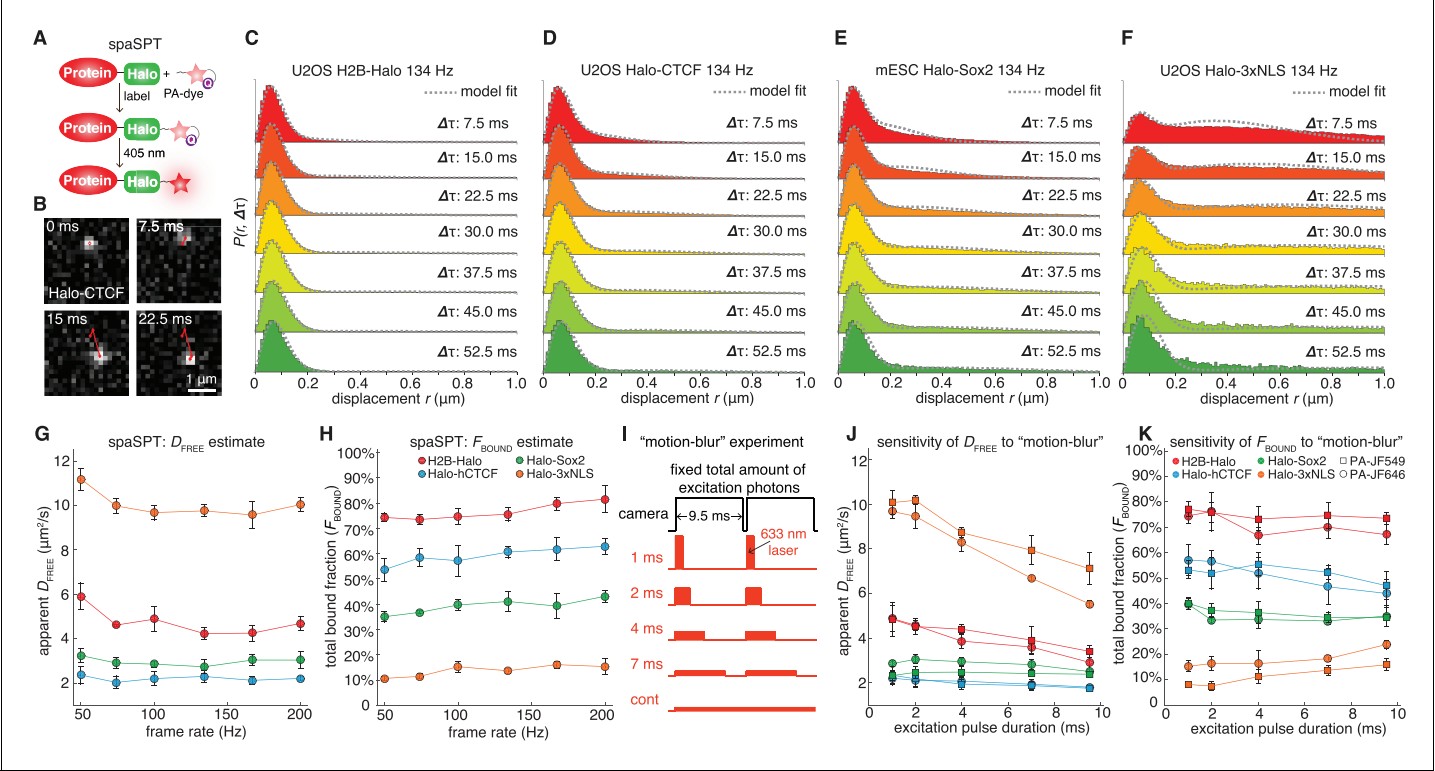

**Figure 4.** Overview of spaSPT and experimental results. (**A**) spaSPT. HaloTag-labeling with UV (405 nm) photo-activatable dyes enable spaSPT. spaSPT minimizes tracking errors through photo-activation which maintains low densities. (**B**) Example data. Raw spaSPT images for Halo-CTCF tracked in human U2OS cells at 134 Hz (1 ms stroboscopic 633 nm excitation of JF$_{646}$). (**C–F**) Histograms of displacements for multiple $\Delta\tau$ of histone H2B-Halo in U2OS cells (**C**), Halo-CTCF in U2OS cells (**d**), Halo-Sox2 in mES cells (**E**) and Halo-3xNLS in U2OS cells (**F**). (**G–H**) Effect of frame-rate on $D_{FREE}$ and $F_{BOUND}$. spaSPT was performed at 200 Hz, 167 Hz, 134 Hz, 100 Hz, 74 Hz and 50 Hz using the 4 cell lines and the data fit using Spot-On and a 2-state model. Each experiment on each cell line was performed in four replicates on different days and ~5 cells imaged each day. (**I**) Motion-blur experiment. To investigate the effect of 'motion-blurring', the total number of excitation photons was kept constant, but delivered during pulses of duration 1, 2, 4, 7 ms or continuous (cont) illumination. (**J–K**) Effect of motion-blurring on $D_{FREE}$ and $F_{BOUND}$. spaSPT data was recorded at 100 Hz and 2-state model-fitting performed with Spot-On. The inferred $D_{FREE}$ (**J**) and $F_{BOUND}$ (**K**) were plotted as a function of excitation pulse duration. Each experiment on each cell line was performed in four replicates on different days and ~5 cells imaged each day. Error bars show standard deviation between replicates.

DOI: https://doi.org/10.7554/eLife.33125.020

The following figure supplements are available for figure 4:

**Figure supplement 1.** Experimental measurement of axial detection range.
DOI: https://doi.org/10.7554/eLife.33125.021

**Figure supplement 2.** Sensitivity of Spot-On to anomalous diffusion.
DOI: https://doi.org/10.7554/eLife.33125.022

**Figure supplement 3.** Re-analysis of experimental data using vbSPT.
DOI: https://doi.org/10.7554/eLife.33125.023

## Effect of motion-blur bias on parameter estimates

Having validated Spot-On on experimental SPT data, we next applied Spot-On to estimate the effect of motion-blurring on the estimation of subpopulations. As mentioned, since most localization algorithms (*Chenouard et al., 2014*; *Sergé et al., 2008*) achieve super-resolution through PSF-fitting, this may cause motion-blurred molecules to be undersampled, resulting in a bias towards slow-moving molecules (*Figure 1A*). We estimated the extent of the bias by imaging the four cell lines at 100 Hz and keeping the total number of excitation photons constant, but varying the excitation pulse duration (1 ms, 2 ms, 4 ms, 7 ms, constant; *Figure 4I*). For generality, we performed these experiments using both PA-JF$_{549}$ and PA-JF$_{646}$ dyes (*Grimm et al., 2016a*). We used Spot-On to fit the data and plotted the apparent free diffusion constant (*Figure 4J*) and apparent total bound fraction (*Figure 4K*) as a function of the excitation pulse duration. For fast-diffusing proteins like Halo-

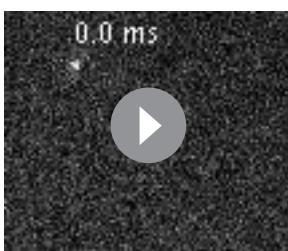

**Video 2.** Related to *Figure 4*. Representative raw spaSPT movie (Halo-hCTCF at 134 Hz). spaSPT movie (1 ms of 633 nm laser delivered at the beginning of each frame; 405 nm laser photo-activation pulses delivered in between frames) of endogenously tagged CTCF (C32 Halo-hCTCF) in human U2OS cells imaged at ~134 Hz (7.477 ms per frame). Dye: PA-JF$_{646}$. One pixel: 160 nm.

DOI: https://doi.org/10.7554/eLife.33125.024

3xNLS and H2B-Halo, motion-blurring resulted in a large underestimate of the free diffusion constant, whereas the effect on slower proteins like CTCF and Sox2 was minor (*Figure 4J*). Regarding the total bound fraction, motion-blurring caused a ~2 fold overestimate for rapidly diffusing Halo-3xNLS (*Figure 4K*), but had a minor effect on slower proteins like H2B, CTCF and Sox2. Similar results were obtained for both dyes for proteins with a significant bound fraction, but we note that JF$_{549}$ appears to better capture the dynamics of proteins with a minimal bound fraction such as Halo-3xNLS (*Figure 4J–K*). Finally, we note that the extent of the bias due to motion-blurring will likely be very sensitive to the localization algorithm. Here, using the MTT-algorithm (*Sergé et al., 2008*), motion-blurring caused up to a 2-fold error in both the $D_{FREE}$ and $F_{BOUND}$ estimates.

Taken together, these results suggest that Spot-On can reliably be used even for SPT data collected under constant illumination provided that protein diffusion is sufficiently slow and, moreover, provides a helpful guide for optimizing SPT imaging acquisitions (we include a full discussion of considerations for SPT acquisitions and a proposal for minimum reporting standards in SPT in Appendix 3 and 4).

## Discussion

In summary, SPT is an increasingly popular technique and has been revealing important new biological insight. However, a clear consensus on how to perform and analyze SPT experiments is currently lacking. In particular, 2D SPT of fast-diffusing molecules inside 3D cells is subject to a number of inherent experimental (*Figure 1A–B*) and analysis (*Figure 1C–D*) biases, which can lead to inaccurate conclusions if not carefully corrected for.

Here, we introduce approaches for accounting for both experimental and analysis biases. Several methods are available for localization/tracking (*Chenouard et al., 2014*; *Sergé et al., 2008*) and for classification of individual trajectories (*Monnier et al., 2015*; *Persson et al., 2013*). Spot-On now complements these tools by providing a bias-corrected, comprehensive open-source framework for inferring subpopulations and diffusion constants from pooled SPT data and makes this platform available through a convenient web-interface. This platform can easily be extended to other diffusion regimes (*Metzler et al., 2014*) and models (*Lee et al., 2017*) and, as 3D SPT methods mature, to 3D SPT data. Moreover, spaSPT provides an acquisition protocol for tracking fast-diffusing molecules with minimal bias. We hope that these validated tools will help make SPT more accessible to the community and contribute positively to the emergence of 'gold-standard' acquisition and analysis procedures for SPT.

## Materials and methods

**Key resources table**

| Reagent type (species) or resource | Designation | Source or reference | Identifiers | Additional information |
|---|---|---|---|---|
| cell line (*Homo sapiens*) | Halo-CTCF | Hansen *et al. eLife* 2017;6:e25776; PMID 28467304; doi: 10.7554/eLife.25776 | U2OS C32 FLAG-Halo-CTCF | Previously reported homozygous endogenous knock-in cell line where all endogenous copies of CTCF have been N-terminally tagged with FLAG-HaloTag |

*Continued on next page*

*Continued*

| Reagent type (species) or resource | Designation | Source or reference | Identifiers | Additional information |
|---|---|---|---|---|
| cell line (*Homo sapiens*) | Halo-3xNLS | Hansen *et al. eLife* 2017;6:e25776; PMID 28467304; doi: 10.7554/eLife.25776 | U2OS Halo-3xNLS | U2OS cell line stably expressing Halo-3xNLS (3 copies of the SV40 Nuclear Localization Signal) generated by G418 selection. Generously provided by David T McSwiggen. |
| cell line (*Homo sapiens*) | H2B-Halo | Hansen *et al. eLife* 2017;6:e25776; PMID 28467304; doi: 10.7554/eLife.25776 | U2OS H2B-Halo-SNAP | U2OS cell line stably expressing histone H2B-Halo-SNAP generated by G418 selection. Generously provided by David T McSwiggen. |
| cell line (*Mus musculus*) | Halo-Sox2 | Teves *et al. eLife* 2016;5:e22280; PMID 27855781; doi: 10.7554/eLife.22280 | mESC JM8.N4 C3 Halo-FLAG-Sox2 | Previously reported homozygous endogenous knock-in cell line where both endogenous copies of Sox2 have been N-terminally tagged with HaloTag-FLAG. Generously provided by Sheila S Teves. |
| software, algorithm | Spot-On Matlab | this paper | Spot-On Matlab | Please see Materials and Methods for a full description. Open-source code is freely available at GitLab: : https://gitlab.com/tjian-da com/elifesciences-publications/spot-on-matlab) |
| software, algorithm | Spot-On Python | this paper | Spot-On Python | Please see Materials and Methods for a full description. Open-source code is freely available at GitLab: https://gitlab.com/tjian-darzac elifesciences-publications/spot-on-cli) |
| software, algorithm | Spot-On | this paper | Spot-On | Please see Materials and Methods for a full description. The web-interface can be found at https://spoton.berkeley.edu/ and the underlying source-code is freely available at GitLab: https://gitlab.com/tjian-darzac elifesciences-publications/spot-on) |
| software, algorithm | simSPT | this paper | simSPT | Code for efficiently simulating experimentally realistic SPT data. Please see Materials and Methods for a full description. Open-source code is freely available at GitLab: https://gitlab.com/tjian-darzacq-lab/simSPT |
| software, algorithm | MSDi; vbSPT; | this paper and Persson *et al. Nature Methods* 2013; PMID: 23396281; DOI: 10.1038/nmeth.2367 | MSDi; vbSPT; | Supplementary software used for MSDi and vbSPT analysis as well as for generating the simulated data can be found at: https://zenodo.org/record/835171 |
| chemical compound, drug | PA-JF$_{549}$ | Grimm *et al. Nature Methods* 2016; PMID 27776112; DOI: 10.1038/nmeth.4034 | PA-JF$_{549}$ | Please contact Luke D Lavis for distribution. |
| chemical compound, drug | PA-JF$_{646}$ | Grimm *et al. Nature Methods* 2016; PMID 27776112; DOI: 10.1038/nmeth.4034 | PA-JF$_{646}$ | Please contact Luke D Lavis for distribution. |

## Spot-On model

Spot-On implements and extends a kinetic modeling framework first described in *Mazza et al. (2012)* and later extended in *Hansen et al. (2017)*. Briefly, the model infers the diffusion constant and relative fractions of two or three subpopulations from the distribution of displacements (or histogram of displacements) computed at increasing lag time ($1\Delta\tau$, $2\Delta\tau$,. ..). This is performed by fitting a semi-analytical model to the empirical histogram of displacements using non-linear least squares fitting. Defocalization is explicitly accounted for by modeling modeling the fraction of particles that remain in focus over time as a function of their diffusion constant.

Mathematically, the evolution over time of a concentration of particles located at the origin as a Dirac delta function and which follows free diffusion in two dimensions with a diffusion constant $D$ can be described by a propagator (also known as Green's function). Properly normalized, the

probability of a particle starting at the origin ending up at a location $r = (x, y)$ after a time delay, $\Delta\tau$, is given by:

$$P(r, \Delta\tau) = N \frac{r}{2D\Delta\tau} e^{\frac{-r^2}{4D\Delta\tau}}$$

Here $N$ is a normalization constant with units of length. Spot-On integrates this distribution over a small histogram bin window, $\Delta r$, to obtain a normalized distribution, the distribution of displacement lengths to compare to binned experimental data. For simplicity, we will therefore leave out $N$ from subsequent expressions. Since experimental SPT data is subject to a significant mean localization error, $\sigma$, Spot-On also accounts for this (**Matsuoka et al., 2009**):

$$P(r, \Delta\tau) = \frac{r}{2(D\Delta\tau + \sigma^2)} e^{\frac{-r^2}{4(D\Delta\tau + \sigma^2)}}$$

Many proteins studied by SPT can generally exist in a quasi-immobile state (e.g. a chromatin-bound state in the case of transcription factors) and one or more mobile states. We will first consider the 2-state model. Under most conditions, state transitions can be ignored ((**Hansen et al., 2017**) and **Figure 3—figure supplement 10**). Thus, the steady-state 2-state model considered by Spot-On becomes:

$$P(r, \Delta\tau) = F_{\text{BOUND}} \frac{r}{2(D_{\text{BOUND}}\Delta\tau + \sigma^2)} e^{\frac{-r^2}{4(D_{\text{BOUND}}\Delta\tau + \sigma^2)}} + (1 - F_{\text{BOUND}}) \frac{r}{2(D_{\text{FREE}}\Delta\tau + \sigma^2)} e^{\frac{-r^2}{4(D_{\text{FREE}}\Delta\tau + \sigma^2)}}$$

Here, the quasi-immobile subpopulation has diffusion constant, $D_{\text{BOUND}}$, and makes up a fraction, $F_{\text{BOUND}}$, whereas the freely diffusing subpopulation has diffusion constant, $D_{\text{FREE}}$, and makes up a fraction, $F_{\text{FREE}} = 1 - F_{\text{BOUND}}$. To account for defocalization bias (**Figure 1C**), Spot-On explicitly considers the probability of the freely diffusing subpopulation moving out of the axial detection range, $\Delta z$, during each time delay, $\Delta\tau$. This is important. For example, only ~25% of freely-diffusing molecules will remain in focus for at least five frames (assuming $\Delta\tau = 10$ ms; $\Delta z$=700 nm; one gap allowed; $D = 5$ μm²/s), resulting in a 4-fold undercounting if uncorrected for. If we assume absorbing boundaries such that any molecule that contacts the edges of the axial detection range located at $z_{\text{MAX}} = \Delta z/2$ and $z_{\text{MIN}} = -\Delta z/2$ is permanently lost, the fraction of freely diffusing molecules with diffusion constant, $D_{\text{FREE}}$, that remain at time delay, $\Delta\tau$, is given by (**Carslow and Jaeger, 1959**; **Kues and Kubitscheck, 2002**):

$$P_{\text{remaining}}(\Delta\tau, \Delta z, D_{\text{FREE}}) = \frac{1}{\Delta z} \int_{-\Delta z/2}^{\Delta z/2} \left\{ 1 - \sum_{n=0}^{\infty} (-1)^n \left[ \text{erfc}\left( \frac{\frac{(2n+1)\Delta z}{2} - z}{\sqrt{4D_{\text{FREE}}\Delta\tau}} \right) + \text{erfc}\left( \frac{\frac{(2n+1)\Delta z}{2} + z}{\sqrt{4D_{\text{FREE}}\Delta\tau}} \right) \right] \right\} dz$$

However, this analytical expression overestimates the fraction lost since there is a significant probability that a molecule that briefly contacted or exceeded the boundary re-enters the axial detection range. The re-entry probability depends on the number of gaps allowed in the tracking ($g$), $\Delta\tau$, and $\Delta z$ and can be approximately accounted for by considering a corrected axial detection range, $\Delta z_{\text{corr}}$, larger than $\Delta z$: $\Delta z_{\text{corr}} > \Delta z$:

$$\Delta z_{\text{corr}}(\Delta z, \Delta\tau, D_{\text{FREE}}, g) = \Delta z + a(\Delta z, \Delta\tau, g)\sqrt{D_{\text{FREE}}} + b(\Delta z, \Delta\tau, g)$$

Although $\Delta z_{\text{corr}}$ depend on the number of gaps ($g$) allowed in the tracking, we will leave it out for simplicity in the following. We determined the coefficients $a$ and $b$ from Monte Carlo simulations. For a given diffusion constant, $D$, 50,000 molecules were randomly placed one-dimensionally along the $z$-axis drawn from a uniform distribution from $z_{\text{MIN}} = -\Delta z/2$ to $z_{\text{MAX}} = \Delta z/2$. Next, using a time-step $\Delta\tau$, one-dimensional Brownian diffusion was simulated along the $z$-axis using the Euler-Maruyama scheme. For time delays from $1\Delta\tau$ to $15\Delta\tau$, the fraction of molecules that were lost was calculated in the range of $D$=[1;12] μm²/s. $a(\Delta z, \Delta\tau, g)$ and $b(\Delta z, \Delta\tau, g)$ were then estimated through least-squares fitting of $P_{\text{remaining}}(\Delta\tau, \Delta z_{\text{corr}}, D)$ to the simulated fraction remaining. The process was repeated over a grid of plausible values of $(\Delta z, \Delta\tau, g)$ to derive a grid of 134,865 $(a, b)$ parameter pairs. This pre-calculated library of $(a, b)$ parameters enables Spot-On to perform model fitting on nearly any SPT dataset with minimal overhead.

Thus, the 2-state model Spot-On uses for kinetic modeling of SPT data is given by:

$$P_2(r, \Delta\tau) = F_{\text{BOUND}} \frac{r}{2(D_{\text{BOUND}}\Delta\tau + \sigma^2)} e^{\frac{-r^2}{4(D_{\text{BOUND}}\Delta\tau + \sigma^2)}}$$
$$+ Z_{\text{CORR}}(\Delta\tau, \Delta z_{\text{corr}}, D_{\text{FREE}})(1 - F_{\text{BOUND}}) \frac{r}{2(D_{\text{FREE}}\Delta\tau + \sigma^2)} e^{\frac{-r^2}{4(D_{\text{FREE}}\Delta\tau + \sigma^2)}}$$

where:

$$Z_{\text{CORR}}(\Delta\tau, \Delta z_{\text{corr}}, D_{\text{FREE}}) = \frac{1}{\Delta z_{\text{corr}}} \int_{-\frac{\Delta z_{\text{corr}}}{2}}^{\frac{\Delta z_{\text{corr}}}{2}} \left\{ 1 - \sum_{n=0}^{\infty} (-1)^n \left[ \text{erfc}\left( \frac{\frac{(2n+1)\Delta z_{\text{corr}}}{2} - z}{\sqrt{4D_{\text{FREE}}\Delta\tau}} \right) + \text{erfc}\left( \frac{\frac{(2n+1)\Delta z_{\text{corr}}}{2} + z}{\sqrt{4D_{\text{FREE}}\Delta\tau}} \right) \right] \right\} dz$$

Having derived the 2-state model, generalization to a 3-state model with 1 bound and 2 diffusive states is straightforward. If the three subpopulations have diffusion constants $D_{\text{BOUND}}$, $D_{\text{SLOW}}$, $D_{\text{FAST}}$, and fractions $F_{\text{BOUND}}$, $F_{\text{SLOW}}$, $F_{\text{FAST}}$, such that $F_{\text{BOUND}} + F_{\text{SLOW}} + F_{\text{FAST}} = 1$, then the 3-state model considered by Spot-On becomes:

$$P_3(r, \Delta\tau) = F_{\text{BOUND}} \frac{r}{2(D_{\text{BOUND}}\Delta\tau + \sigma^2)} e^{\frac{-r^2}{4(D_{\text{BOUND}}\Delta\tau + \sigma^2)}}$$
$$+ Z_{\text{CORR}}(\Delta\tau, \Delta z_{\text{corr}}, D_{\text{SLOW}}) F_{\text{SLOW}} \frac{r}{2(D_{\text{SLOW}}\Delta\tau + \sigma^2)} e^{\frac{-r^2}{4(D_{\text{SLOW}}\Delta\tau + \sigma^2)}}$$
$$+ Z_{\text{CORR}}(\Delta\tau, \Delta z_{\text{corr}}, D_{\text{FAST}})(1 - F_{\text{BOUND}} - F_{\text{SLOW}}) \frac{r}{2(D_{\text{FAST}}\Delta\tau + \sigma^2)} e^{\frac{-r^2}{4(D_{\text{FAST}}\Delta\tau + \sigma^2)}}$$

Where $Z_{\text{CORR}}(\Delta\tau, \Delta z_{\text{corr}}, D)$ is as described above.

## Numerical implementation of models in Spot-On

Spot-On calculates the empirical histogram of displacements based on a user-defined bin width. Spot-On allows the user to choose between PDF- and CDF-fitting of the kinetic model to the empirical displacement distributions; CDF-fitting is generally most accurate for smaller datasets and the two are similar for large datasets (*Figure 3—figure supplement 9*). The integral in $Z_{\text{CORR}}(\Delta\tau, \Delta z_{\text{corr}})$ was numerically evaluated using the midpoint method over 200 points and the terms of the series computed until the term falls below a threshold of $10^{-10}$. Model fitting and parameter optimization was performed using a non-linear least squares algorithm (Levenberg-Marquardt). Random initial parameter guesses are drawn uniformly from the user-specified parameter range. The optimization is then repeated several times with different initialization parameters to avoid local minima. Spot-On constrains each fraction to be between 0 and 1 and for the sum of the fractions to equal 1.

## Theoretical characteristics and limitations of the model

Although Spot-On performs well on both experimental and simulated SPT data, the model implemented by Spot-On has several limitations. First, the kinetic model assumes diffusion to be *ideal Brownian motion*, even though it is widely acknowledged that the motion of most proteins inside a cell shows some degree of anomalous diffusion. Nevertheless, *Figure 4G–H* and *Figure 4—figure supplement 2* show that the parameter inference for experimental data of proteins presenting various degrees of anomalous diffusion is quite robust.

Second, Spot-On models the localization error as the *static mean localization error* and this feature can be used to infer the actual localization error from the data. However, the localization error is affected both by the position of the particle with respect to the focal plane (*Lindén et al., 2017*) and by motion blur (*Deschout et al., 2012*). Even though a high signal-to-background ratio and fast framerate/stroboscopic illumination help to mitigate these disparities, it is likely that the localization error of fast moving particles will be higher than the bound/slow-moving particles. In that case, one would expect Spot-On to infer a localization error that is the weighted mean of the 'bound/static' localization error and the 'free' localization error. However, in many situations $D_{\text{free}}\Delta\tau \gg \sigma^2$ (even assuming a 2 $\mu m^2$/s particle imaged at a 5 ms framerate with a ~30 nm localization error, there is still an order of magnitude difference between the two terms). As a consequence, the estimate of $\sigma$ reflects the static localization error (that is, the localization error of the bound fraction), and the localization error estimate becomes less reliable if the bound fraction is very small (*Figure 3—figure supplement 11*).

Third, following (*Kues and Kubitscheck, 2002*) the *axial detection profile* is assumed to be a step function, which is an approximation. However, all simulations here were performed using a detection

profile with Gaussian edges (*Figure 3—figure supplement 1*) and as shown in *Figure 3A–B* Spot-On still works quite well and moreover is relatively robust to slight mismatches in the axial detection range (*Figure 3—figure supplement 7*).

Fourth, unlike the original implementation by *Mazza et al. (2012)*, Spot-On ignores *state transitions*. This reduces the number of fitted parameters and simplifies the generalization to more than two states, but as shown in *Figure 3—figure supplement 10* it also causes the parameter inference to fail unless the timescale of state changes is at least 10–50 times longer than the frame rate. Thus, in cases where a molecule is known to exhibit state changes on a time-scale of tens to a few hundreds of milliseconds, Spot-On may not be appropriate.

Fifth and finally, Spot-On ignores correlations between adjacent displacements, although taking such information into account can potentially improve the parameter inference (*Vestergaard et al., 2014*).

## Cell culture

Halo-Sox2 (*Teves et al., 2016*) knock-in JM8.N4 mouse embryonic stem cells ((*Pettitt et al., 2009*) Research Resource Identifier: RRID:CVCL_J962; obtained from the KOMP Repository at UC Davis) were grown on plates pre-coated with a 0.1% autoclaved gelatin solution (Sigma-Aldrich, St. Louis, MO, G9391) under feeder free conditions in knock-out DMEM with 15% FBS and LIF (full recipe: 500 mL knockout DMEM (ThermoFisher, Waltham, MA, #10829018), 6 mL MEM NEAA (ThermoFisher #11140050), 6 mL GlutaMax (ThermoFisher #35050061), 5 mL Penicillin-streptomycin (ThermoFisher #15140122), 4.6 µL 2-mercapoethanol (Sigma-Aldrich M3148), 90 mL fetal bovine serum (HyClone Logan, UT, FBS SH30910.03 lot #AXJ47554)) and LIF. mES cells were fed by replacing half the medium with fresh medium daily and passaged every two days by trypsinization. Halo-3xNLS, H2B-Halo-SNAP and knock-in C32 Halo-CTCF (*Hansen et al., 2017*) Human U2OS osteosarcoma cells (Research Resource Identifier: RRID:CVCL_0042) were grown in low glucose DMEM with 10% FBS (full recipe: 500 mL DMEM (ThermoFisher #10567014), 50 mL fetal bovine serum (HyClone FBS SH30910.03 lot #AXJ47554) and 5 mL Penicillin-streptomycin (ThermoFisher #15140122)) and were passaged every 2–4 days before reaching confluency. For live-cell imaging, the medium was identical except DMEM without phenol red was used (ThermoFisher #31053028). Both mouse ES and human U2OS cells were grown in a Sanyo copper alloy IncuSafe humidified incubator (MCO-18AIC(UV)) at 37°C/5.5% $CO_2$. Cell lines were pathogen tested and authenticated through STR profiling (U2OS) as described previously (*Hansen et al., 2017*; *Teves et al., 2016*). All cell lines will be provided upon request.

## Single-molecule imaging

The indicated cell line was grown overnight on plasma-cleaned 25 mm circular no 1.5H cover glasses (Marienfeld, Germany, High-Precision 0117650) either directly (U2OS) or MatriGel coated (mESCs; Fisher Scientific, Hampton, NH, #08-774-552 according to manufacturer's instructions just prior to cell plating). After overnight growth, cells were labeled with 5–50 nM PA-JF$_{549}$ or PA-JF$_{646}$ (*Grimm et al., 2016a*) for ~15–30 min and washed twice (one wash: medium removed; PBS wash; replenished with fresh medium). At the end of the final wash, the medium was changed to phenol red-free medium keeping all other aspects of the medium the same. Single-molecule imaging was performed on a custom-built Nikon TI microscope (Nikon Instruments Inc., Melville, NY) equipped with a 100x/NA 1.49 oil-immersion TIRF objective (Nikon apochromat CFI Apo TIRF 100x Oil), EM-CCD camera (Andor, Concord, MA, iXon Ultra 897; frame-transfer mode; vertical shift speed: 0.9 µs; −70°C), a perfect focusing system to correct for axial drift and motorized laser illumination (Ti-TIRF, Nikon), which allows an incident angle adjustment to achieve highly inclined and laminated optical sheet illumination (*Tokunaga et al., 2008*). The incubation chamber maintained a humidified 37°C atmosphere with 5% $CO_2$ and the objective was also heated to 37°C. Excitation was achieved using the following laser lines: 561 nm (1 W, Genesis Coherent, Santa Clara, CA) for PA-JF$_{549}$; 633 nm (1 W, Genesis Coherent, Pala Alto, CA) for PA-JF$_{646}$; 405 nm (140 mW, OBIS, Coherent) for all photo-activation experiments. The excitation lasers were modulated by an acousto-optic Tunable Filter (AA Opto-Electronic, France, AOTFnC-VIS-TN) and triggered with the camera TTL exposure output signal. The laser light is coupled into the microscope by an optical fiber and then reflected using a multi-band dichroic (405 nm/488 nm/561 nm/633 nm quad-band, Semrock, Rochester, NY) and then

focused in the back focal plane of the objective. Fluorescence emission light was filtered using a single band-pass filter placed in front of the camera using the following filters: PA-JF549: Semrock 593/40 nm bandpass filter; PA-JF$_{646}$: Semrock 676/37 nm bandpass filter. The microscope, cameras, and hardware were controlled through NIS-Elements software (Nikon).

## spaSPT experiments and analysis

The spaSPT experimental settings for *Figure 4G–H* were as follows: 1 ms 633 nm excitation (100% AOTF) of PA-JF$_{646}$ was delivered at the beginning of the frame; 405 nm photo-activation pulses were delivered during the camera integration time (~447 μs) to minimize background and their intensity optimized to achieve a mean density of ≤1 molecule per frame per nucleus. 30,000 frames were recorded per cell per experiment. The camera exposure times were: 4.5 ms, 5.5 ms, 7 ms, 9.5 ms, 13 ms and 19.5 ms.

For the motion-blur spaSPT experiments (*Figure 4I–K*), the camera exposure was fixed to 9.5 ms and photo-activation performed as above. To keep the total number of delivered photons constant, we generated an AOTF-laser intensity calibration curve using a power meter and adjusted the AOTF transmission accordingly for each excitation pulse duration. The excitation settings were as follows: 1 ms, 561 nm 100% AOTF, 633 nm 100% AOTF; 2 ms, 561 nm 43% AOTF, 633 nm 40% AOTF; 4 ms, 561 nm 28% AOTF, 633 nm 27% AOTF; 7 ms, 561 nm 20% AOTF, 633 nm 19% AOTF; constant illumination, 561 nm 17% AOTF, 633 nm 16% AOTF.

spaSPT data was analyzed (localization and tracking) and converted into trajectories using a custom-written Matlab implementation of the MTT-algorithm (*Sergé et al., 2008*) and the following settings: Localization error: $10^{-6.25}$; deflation loops: 0; Blinking (frames): 1; max competitors: 3; max $D$ (μm$^2$/s): 20. The spaSPT trajectory data was then analyzed using the Matlab version of Spot-On (v1.0; GitLab tag 1f9f782b) and the following parameters: dZ = 0.7 μm; GapsAllowed = 1; Time-Points: 4 (50 Hz), 6 (74 Hz), 7 (100 Hz), 8 (134 Hz), 9 (167 and 200 Hz); JumpsToConsider = 4; Model-Fit = 2; NumberOfStates = 2; FitLocError = 0; LocError = 0.035 μm; D_Free_2State=[0.4;25]; D_Bound_2State=[0.00001;0.08];

## SPT simulations

We developed a utility to simulate diffusing proteins in a confined geometry (simSPT). Briefly, simSPT simulates the diffusion of an arbitrary number of populations of molecules characterized by their diffusion coefficient, under a steady state assumption. Particles are drawn at random between the populations and their location in the 3D nucleus is initialized following a uniform law within the confinement volume. The lifetime of the particle (in frames) is also drawn following an exponential law of mean lifetime $\beta$. Then, the particle diffuses in 3D until it bleaches. Diffusion is simulated by drawing jumps following a normal law of parameters $N(0, \sqrt{2D\Delta\tau})$, where $D$ is the diffusion coefficient and $\Delta\tau$ the exposure time. Finally, a localization error ($N(0, \sigma)$) is added to each $(x,y,z)$ localization in the simulated trajectories.

For comparisons of Spot-On, MSD$_i$ and vbSPT using a 2-state scenario, we parameterized simSPT to consider two subpopulations of particles diffusing in a sphere (the nucleus) of 8 μm diameter illuminated using HiLo illumination (assuming a HiLo beam width of 4 μm), with an axial detection range of ~700 nm, centered at the middle of the HiLo beam with Gaussian edges. Molecules are assumed to have a mean lifetime of 4 frames (when inside the HiLo beam) and of 40 frames when outside the HiLo beam. The localization error was set to 25 nm and the simulation was run until 100,000 in-focus trajectories were recorded. More specifically, the effect of the exposure time (1 ms, 4 ms, 7 ms, 13 ms, 20 ms), the free diffusion constant (from 0.5 μm$^2$/s to 14.5 μm$^2$/s in 0.5 μm$^2$/s increments) and the fraction bound (from 0% to 95% in 5% increments) were investigated, yielding a dataset consisting of 3480 simulations. More details on the simulations, including scripts to reproduce the dataset, are available on GitLab as detailed in the 'Computer code' section. Full details on how the simulations were analyzed by Spot-On, vbSPT and MSD$_i$ are given in Appendix 1.

We also considered a 3-state scenario featuring a bound subpopulation ('bound'), a relatively slow diffusing free subpopulation ('slow') and a relatively faster diffusing free subpopulation ('free'). In this case, we only compared Spot-On and vbSPT (*Figure 3—figure supplement 6*), since the MSD$_i$ methods did not perform well. As in the 2-state simulations, we parameterized simSPT to consider that three subpopulations of particles diffusing in a sphere (the nucleus) of 8 μm diameter

illuminated using HiLo illumination (assuming a HiLo beam width of 4 µm), with an axial detection range of ~700 nm, centered at the middle of the HiLo beam with Gaussian edges. Molecules are assumed to have a mean lifetime of 4 frames (when inside the HiLo beam) and of 40 frames when outside the HiLo beam. The localization error was set to 40 nm and the simulation was run until 100,000 in-focus trajectories were recorded. We considered three different subpopulation conditions: (1) $F_{BOUND}$ = 25%; $F_{SLOW}$ = 25%; $F_{FAST}$ = 50%; (2) $F_{BOUND}$ = 25%; $F_{SLOW}$ = 50%; $F_{FAST}$ = 25%; (3) $F_{BOUND}$ = 50%; $F_{SLOW}$ = 25%; $F_{FAST}$ = 25%. Specifically, for each of these condition, the effect of of the exposure time (1 ms, 4 ms, 7 ms, 10 ms, 13 ms, 20 ms), the slower free diffusion constant (from 0.5 µm²/s to 2.5 µm²/s in 0.5 µm²/s increments) and the faster free diffusion constant (from 4 µm²/s to 11 µm²/s in 1 µm²/s increments) were investigated, yielding a dataset of 720 simulations. Both vbSPT and Spot-On (all) were constrained to three subpopulations. Full details on how the simulations were analyzed by Spot-On and vbSPT are given in Appendix 1.

## Data availability

All raw 1064 spaSPT experiments (*Figure 4*) as well as the 3480 simulations (*Figure 3*) are freely available in Spot-On readable Matlab and CSV file formats in the form of SPT trajectories at Zenodo. The experimental data is available at: https://zenodo.org/record/834781; The simulations are available in Matlab format at: https://zenodo.org/record/835541; The simulations are available in CSV format at: https://zenodo.org/record/834787; And supplementary software used for MSD$_i$ and vbSPT analysis as well as for generating the simulated data at: https://zenodo.org/record/835171

## Computer code

Spot-On is fully open-source. The web-interface can be found at: https://SpotOn.berkeley.edu. All raw code is available at GitLab: https://gitlab.com/tjian-darzacq-lab. The web-interface code can be found at https://gitlab.com/tjian-darzacq-lab/Spot-On; the Matlab command-line version of Spot-On can be found at: https://gitlab.com/tjian-darzacq-lab/spot-on-matlab; the Python command-line version of Spot-On can be found at https://gitlab.com/tjian-darzacq-lab/Spot-On-cli; the SPT simulation code (simSPT) can be found at: https://gitlab.com/tjian-darzacq-lab/simSPT; finally, the 'TrackMate to Spot-On connector' plugin, which adds an extra menu to TrackMate which allows one-click upload of datasets to Spot-On can be found at: https://gitlab.com/tjian-darzacq-lab/Spot-On-TrackMate

## Acknowledgements

ASH and MW contributed equally to this work and are alphabetically listed. We are very grateful to Davide Mazza who inspired this work and provided invaluable comments on Spot-On, to Florian Mueller for suggestions on the web-application, Christophe Zimmer for insightful discussions, David McSwiggen and Sheila Teves for kindly providing cell lines, Carolyn Elya and Chiahao Tsui for the name 'Spot-On', and to members of the Tjian/Darzacq labs and Maxime Dahan for discussions. We also thank Astou Tangara and Anatalia Robles for microscope maintenance. ASH is a postdoctoral fellow of the Siebel Stem Cell Institute. This work was supported by NIH grants UO1-EB021236 and U54-DK107980 (XD), the California Institute of Regenerative Medicine grant LA1-08013 (XD), by the Howard Hughes Medical Institute (003061, RT) and used the computational and storage services (TARS cluster) provided by the IT department at Institut Pasteur, Paris.

## Additional information

### Competing interests

Jonathan B Grimm, Luke D Lavis: has filed patent applications (e.g. PCT/US2015/023953) whose value may be affected by this publication. Robert Tjian: One of the three founding funders of eLife and a member of eLife's Board of Directors. The other authors declare that no competing interests exist.

## Funding

| Funder | Grant reference number | Author |
|---|---|---|
| National Institutes of Health | UO1-EB021236 | Xavier Darzacq |
| National Institutes of Health | U54-DK107980 | Xavier Darzacq |
| California Institute for Regenerative Medicine | LA1-08013 | Xavier Darzacq |
| Howard Hughes Medical Institute | 003061 | Robert Tjian |
| Howard Hughes Medical Institute | | Luke D Lavis |
| Siebel Stem Cell Institute | | Anders S Hansen |

The funders had no role in study design, data collection and interpretation, or the decision to submit the work for publication.

## Author contributions

Anders S Hansen, Conceived of the project, Developed Spot-On, Performed simulations, Performed experiments, Analyzed experiments, Drafted and edited the manuscript; Maxime Woringer, Conceived of the project, Developed Spot-On, Performed simulations, Analyzed experiments, Drafted and edited the manuscript; Jonathan B Grimm, Luke D Lavis, Developed and contributed JF dyes, Edited the manuscript; Robert Tjian, Supervised the research, Reviewed and edited the manuscript; Xavier Darzacq, Conceived of the project, Supervised the research, Reviewed and edited the manuscript

## Author ORCIDs

Anders S Hansen http://orcid.org/0000-0001-7540-7858
Maxime Woringer https://orcid.org/0000-0003-2581-9808
Robert Tjian https://orcid.org/0000-0003-0539-8217
Xavier Darzacq http://orcid.org/0000-0003-2537-8395

## Decision letter and Author response

Decision letter https://doi.org/10.7554/eLife.33125.041
Author response https://doi.org/10.7554/eLife.33125.042

# Additional files

## Supplementary files

• Supplementary file 1. PDF of step-by-step manual for using Spot-On.
DOI: https://doi.org/10.7554/eLife.33125.025

• Transparent reporting form
DOI: https://doi.org/10.7554/eLife.33125.026

## Major datasets

The following datasets were generated:

| Author(s) | Year | Dataset title | Dataset URL | Database, license, and accessibility information |
|---|---|---|---|---|
| Hansen AS, Woringer M, Grimm JB, Lavis LD, Tjian R, Darzacq X | 2017 | Experimental data for "Spot-On: robust model-based analysis of single-particle tracking experiments" | https://doi.org/10.5281/zenodo.834781 | Publicly available at Zenodo (https://zenodo.org/) |
| Hansen AS, Woringer M, Grimm JB, Lavis LD, Tjian R, Darzacq X | 2017 | Simulated data for "Spot-On: robust model-based analysis of single-particle tracking experiments" (MATLAB format) | https://doi.org/10.5281/zenodo.835541 | Publicly available at Zenodo (https://zenodo.org/) |

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

## Appendix 1

DOI: https://doi.org/10.7554/eLife.33125.027

## Fitting of simulations using Spot-On, vbSPT and MSD$_i$

To systematically evaluate the performance of Spot-On as well as other common analysis tools such as MSD$_i$ and vbSPT (**Persson et al., 2013**), we developed simSPT, a simulation tool to generate a comprehensive set of realistic SPT simulations spanning the range of plausible dynamics (almost a billion trajectories were simulated in total). simSPT is freely available at GitLab: https://gitlab.com/tjian-darzacq-lab/simSPT. simSPT simulates 3D SPT trajectories arising from an arbitrary number of subpopulations confined inside a sphere under HiLo illumination and takes into account a limited axial detection range, realistic photobleaching rates and optionally state interconversion. The simulation methods are described in detail at GitLab.

Briefly, we parameterized simSPT to consider that particles diffuse inside a sphere (the nucleus) of 8 μm diameter illuminated using HiLo illumination (assuming a HiLo beam width of 4 μm), with an axial detection range of ~700 nm with Gaussian edges, centered at the middle of the HiLo beam. Molecules are assumed to have a mean lifetime of 4 frames (when inside the HiLo beam) and of 40 frames when outside the HiLo beam.

For the 2-state comparisons, the localization error was set to 25 nm and the simulation was run until 100,000 in-focus trajectories were recorded. More specifically, the effect of the time between frames (1 ms, 4 ms, 7 ms, 13 ms, 20 ms), the free diffusion constant (from 0.5 μm$^2$/s to 14.5 μm$^2$/s in 0.5 μm$^2$/s increments) and the fraction bound (from 0% to 95% in 5% increments) were investigated, yielding a dataset consisting of 3480 simulations. All 3480 simulated datasets are also available (see Data Availability section). The advantage of simulations is that the ground truth is known.

For the 3-state comparisons (**Figure 3—figure supplement 6**), the localization error was set to 40 nm and the simulation was run until 100,000 in-focus trajectories were recorded. We then simulated one bound state ($D_{BOUND}$=0.001 μm$^2$/s) and two free states ($D_{SLOW}$=0.5 to 2.5 μm$^2$/s in 0.5 μm$^2$/s increments; $D_{FAST}$= 4.0 to 11.0 μm$^2$/s in 1.0 μm$^2$/s increments) and also varying the fractions ($F_{BOUND}$=25%, $F_{SLOW}$=25%, $F_{FAST}$= 50%; or $F_{BOUND}$=25%, $F_{SLOW}$=50%, $F_{FAST}$= 25%; or $F_{BOUND}$=50%, $F_{SLOW}$=25%, $F_{FAST}$= 25%;) as was the time between frames (1 ms, 4 ms, 7 ms, 10, 13 ms, 20 ms).

For more specific simulations, extra parameters were varied, such as the width of the axial detection range (**Figure 3—figure supplement 7**), localization error (**Figure 3—figure supplement 11**), or the presence/absence of interconversion between states (**Figure 3—figure supplement 10**).

### Comparison of methods for 2-state simulations

In the case of the main 3480 simulated SPT datasets for the 2-state comparison, we analyzed the data using the Matlab version of Spot-On (either using JumpsToConsider = 4 or all), MSD$_i$ (either $R^2$ >0.8 or all) or vbSPT. We describe the analysis in details below.

### Spot-On (4 jumps)

#### Rational and parameters

Spot-On allows a user to use the entirety of each trajectory or to use only the first *n* jumps by adjusting the parameter, *JumpsToConsider*. For example, consider a trajectory consisting of 6 localizations and without gaps. If JumpsToConsider = 4 and TimePoints = 6, then this trajectory will contribute four displacements to the 1Δτ histogram, four displacements to the 2Δτ histogram, three displacements to the 3Δτ histogram, two displacements to the 4Δτ histogram and one displacement to the 5Δτ histogram. Thus, even though the trajectory

contains 5 $1\Delta\tau$ displacements, only the first four will be used for analysis if JumpsToConsider = 4. While on simulated data, using a subset of the trajectories is always slightly less accurate than using the entire trajectory since it slightly underestimates the bound fraction, we previously (*Hansen et al., 2017*) used this as an empirical way of compensating for all the other experimental biases that cause undercounting of freely diffusing molecules that cannot fully be taken into account in simulations. We therefore also tested this approach in the simulations. To fit the simulations using Spot-On we fed the following parameters to the function SpotOn_core.m (v1.0; GitLab tag 1f9f782b):

- dZ = 0.700;
- GapsAllowed = 1;
- BinWidth = 0.010;
- UseAllTraj = 0;
- JumpsToConsider = 4;
- MaxJump = 6.05;
- ModelFit = 2;
- DoSingleCellFit = 0;
- NumberOfStates = 2;
- FitIterations = 2;
- FitLocError = 0;
- LocError = 0.0247;
- D_Free_2State = [0.4 25];
- D_Bound_2State = [0.00001 0.08];
- TimePoints: 10 if 1 ms; 9 if 4 ms; 8 if 7 ms; 7 if 10 ms; 6 if 13 ms; 5 if 20 ms;
- The empirical a,b parameters used to correct for defocalization bias were as follows:
  - $\Delta\tau$ = 1 ms; $\Delta z$ = 0.7 μm; 1 gap: a = 0.0387 $s^{12}$; b = 0.3189 μm;
  - $\Delta\tau$ = 4 ms; $\Delta z$ = 0.7 μm; 1 gap: a = 0.1472 $s^{1/2}$; b = 0.2111 μm;
  - $\Delta\tau$ = 7 ms; $\Delta z$ = 0.7 μm; 1 gap: a = 0.1999 $s^{1/2}$; b = 0.2058 μm;
  - $\Delta\tau$ = 10 ms; $\Delta z$ = 0.7 μm; 1 gap: a = 0.2379 $s^{1/2}$; b = 0.2017 μm;
  - $\Delta\tau$ = 13 ms; $\Delta z$ = 0.7 μm; 1 gap: a = 0.2656 $s^{1/2}$; b = 0.2118 μm;
  - $\Delta\tau$ = 20 ms; $\Delta z$ = 0.7 μm; 1 gap: a = 0.3133 $s^{1/2}$; b = 0.2391 μm;

CDF-fitting was then performed in MATLAB R2014b using the Matlab version of Spot-On (v1.0; GitLab tag 1f9f782b) and the estimated free diffusion constant, $D_{FREE}$, and bound fraction, $F_{BOUND}$, recorded for each of the 3480 simulations. The estimated $D_{FREE}$ and $F_{BOUND}$ were then compared to the ground truth known from the simulations. Three parameters were estimated in the fit.

## Performance evaluation

Spot-On (4 jumps) performs slightly worse than Spot-On (all) when it comes to estimating $F_{BOUND}$ as expected and essentially identically to Spot-On (all) for estimating $D_{FREE}$. The mean error (bias) for estimating $F_{BOUND}$ was −6.4%, the inter-quartile range (IQR) was 5.9% and the standard deviation 3.6%. The origin of the error is the undercounting of the bound population due to considering only the first 4 jumps. Since bound molecules remain in focus until they bleach, they always yield only a single trajectory, whereas a single freely diffusing molecule has a probability of yielding multiple trajectories by diffusing in-focus for a while, then moving out-of-focus for a while and then moving back in-focus. For estimating $D_{FREE}$ the bias for Spot-On (4 jumps) was −5.4%, the IQR 3.6% and the standard deviation 3.2%. However, as shown in *Figure 3—figure supplements 2* and *4*, the slight underestimate of the free diffusion constant is not due to a limitation of Spot-On, but instead due to confinement inside the nucleus (*Figure 3—figure supplement 4*). For example, a diffusing molecule close to the nuclear boundary moving towards the nuclear boundary will 'bounce back' resulting in a large distance travelled, but only a smaller recorded displacement. We validated that this indeed is the origin of the underestimate of $D_{FREE}$ by considering a nucleus with virtually no confinement (20 $\mu$m radius) and found that the $D_{FREE}$-underestimate was now minimal (*Figure 3—figure supplement 4*). Finally, Spot-On always estimated the bound diffusion constant, $D_{BOUND}$, with minimal error unlike $MSD_i$ or vbSPT, which were not able to accurately estimate $D_{BOUND}$. However, since there is generally less interest in $D_{BOUND}$, we did not use this further for evaluating the performance of the different methods.

### Spot-On (all)

**Rational and parameters**: Spot-On (all) was run on the simulations identically to Spot-On (4 jumps) except the entirety of each trajectory was used for calculating the histograms. To fit the simulations using Spot-On we fed the following parameters to the function SpotOn_core.m (v1.0; GitLab tag 1f9f782b):

- dZ = 0.700;
- GapsAllowed = 1;
- BinWidth = 0.010;
- UseAllTraj = 1;
- MaxJump = 6.05;
- ModelFit = 2;
- DoSingleCellFit = 0;
- NumberOfStates = 2;
- FitIterations = 2;
- FitLocError = 0;
- LocError = 0.0247;
- D_Free_2State = [0.4 25];
- D_Bound_2State = [0.00001 0.08];
- TimePoints: 10 if 1 ms; 9 if 4 ms; 8 if 7 ms; 7 if 10 ms; 6 if 12 ms; 5 if 20 ms;
- The empirical a,b parameters used to correct for defocalization bias were as follows:
  - o $\Delta\tau$ = 1 ms; $\Delta z$ = 0.7 µm; 1 gap: a = 0.0387 $s^{1/2}$; b = 0.3189 µm;
  - o $\Delta\tau$ = 4 ms; $\Delta z$ = 0.7 µm; 1 gap: a = 0.1472 $s^{1/2}$; b = 0.2111 µm;
  - o $\Delta\tau$ = 7 ms; $\Delta z$ = 0.7 µm; 1 gap: a = 0.1999 $s^{1/2}$; b = 0.2058 µm;
  - o $\Delta\tau$ = 10 ms; $\Delta z$ = 0.7 µm; 1 gap: a = 0.2379 $s^{1/2}$; b = 0.2017 µm;
  - o $\Delta\tau$ = 13 ms; $\Delta z$ = 0.7 µm; 1 gap: a = 0.2656 $s^{1/2}$; b = 0.2118 µm;
  - o $\Delta\tau$ = 20 ms; $\Delta z$ = 0.7 µm; 1 gap: a = 0.3133 $s^{1/2}$; b = 0.2391 µm;

As above, CDF-fitting was performed and the $D_{FREE}$-estimate and $F_{BOUND}$-estimate compared to the ground truth for each of the 3480 simulations for which the ground truth is known. Three parameters were estimated in the fit.

### Performance evaluation

Spot-On (all) out-performed all other approaches. The mean error (bias) for estimating $F_{BOUND}$ was −1.7%, the inter-quartile range (IQR) was 1.8% and the standard deviation 1.2%. For estimating $D_{FREE}$ the bias for Spot-On (all) was −4.8%, the IQR 3.5% and the standard deviation 3.3%. But as mentioned above, the slight underestimate of $D_{FREE}$ is simply due to diffusion being confined inside a 4 $\mu$m radius nucleus (*Figure 3—figure supplement 4*). This also helps to emphasize the point that diffusion constants measured inside a nucleus should be interpreted as apparent diffusion constants.

### MSD$_i$ ($R^2$>0.8)

### Rational and parameters

A large number of papers have use different variations of the MSD$_i$ approach (*Knight et al., 2015*; *Li et al., 2016*; *Liu et al., 2014*; *Schmidt et al., 2016*; *Zhen et al., 2016*). This approach is of course very sensitive to how the MSD is estimated. For example, it is well-known that accurately estimating diffusion constants from short trajectories (<100 frames) subject to significant localization error is all but impossible as shown by Michalet and Berglund (*Michalet and Berglund, 2012*). Nevertheless, several papers assign diffusion constants to individual trajectories based on a MSD-fit. While the exact method differs somewhat from paper to paper, the most popular approach is to set a threshold of a certain number of localizations per trajectory (most commonly 5; though we note that some reports explicitly attempt to compensate for the bias introduced by setting such a threshold (*Zhen et al., 2016*)). Each trajectory with at least five localizations are then fit, often using the Matlab library MSDAnalyzer (*Tarantino et al., 2014*), and thus assigned an apparent diffusion

constant. An additional threshold is then applied: only if the fit to the MSD curve is judged sufficiently good, is the diffusion constant then used. Otherwise the trajectory is ignored. This fitting threshold is frequently set based on the coefficient of determination as $R^2 > 0.8$ in some recent papers (**Knight et al., 2015**; **Li et al., 2016**; **Schmidt et al., 2016**). Next, after analyzing all trajectories in this way, a distribution of diffusion constants is then obtained. The analysis is then performed on the logarithm of these diffusion constants ('LogD histogram') (**Knight et al., 2015**; **Li et al., 2016**; **Schmidt et al., 2016**). Both the CDF (**Knight et al., 2015**) and PDF (**Knight et al., 2015**; **Li et al., 2016**; **Schmidt et al., 2016**; **Zhen et al., 2016**) can be considered. These are then fitted with a sum of Gaussian distributions: either two (**Knight et al., 2015**; **Schmidt et al., 2016**; **Zhen et al., 2016**) or three (**Schmidt et al., 2016**; **Zhen et al., 2016**). We note that it is not immediately clear which distribution fitted diffusion constants should actually follow (e.g. Log-normal, Gamma, Normal, etc.). No justification is given for sums of Gaussians (**Knight et al., 2015**; **Li et al., 2016**; **Schmidt et al., 2016**), though we note that the fit is often quite good both in the previous reports (**Knight et al., 2015**; **Li et al., 2016**; **Schmidt et al., 2016**) and also here as shown in **Figure 3—figure supplement 3**. Please note that fitting a sum of normal distributions to the LogD histogram is equivalent to fitting a sum of log-normal distributions to the D histogram. We also note here, that in a theoretical study Michalet previously showed that the distribution of diffusion constants is approximately Gaussian, but only under a set of stringent criteria (**Michalet, 2010**). Since CDF-fitting is generally less susceptible to noise from binning and since in this comparison Spot-On also uses CDF-fitting, we fit the LogD histogram with a sum of 2 Gaussians using CDF-fitting. We refer to this whole procedure as $MSD_i$ ($R^2 > 0.8$). Examples of fits are shown in **Figure 3** and **Figure 3—figure supplement 3** and the Matlab code to perform the fitting is available together with the data (see "Data availability"). Five parameters were estimated in the fit.

## Performance evaluation

Overall, $MSD_i$ ($R^2 > 0.8$) generally performs reasonably well when it comes to estimating $D_{FREE}$, but extremely poorly when it comes to $F_{BOUND}$ and $D_{BOUND}$. The mean error (bias) for estimating $D_{FREE}$ was 8.0%, the inter-quartile range (IQR) was 4.9% and the standard deviation 28.5%. For estimating $F_{BOUND}$ the bias for $MSD_i$ ($R^2 > 0.8$) was −20.6%, the IQR 32.1% and the standard deviation 26.4%. We note that since $F_{BOUND}$ necessarily has to take a value between 0% and 95% in the simulations and since half the simulations have $F_{BOUND} < 50\%$, a mean error of −20.6% is actually quite large. Although the bias for $D_{FREE}$ is much smaller, in ~5% of all cases, the error in estimating $D_{FREE}$ is bigger than 2-fold. Moreover, in a few very rare cases, not a single trajectory out of the 100,000 simulated trajectories pass both thresholds ($R^2 > 0.8$; at least five frames). Why is $MSD_i$ ($R^2 > 0.8$) fitting so unreliable? It is instructive to consider an example. In the example dataset provided with the $MSD_i$ code (simulation with $D_{FREE} = 2$; $F_{BOUND} = 0.75$; 1 ms frame rate), the estimated $D_{FREE} = 2.06$ is very good, but the estimated $F_{BOUND} = 0.16$ is extremely poor. Even though the simulation dataset contains 100,000 simulated trajectories, only 3726 of them actually pass the threshold ($R^2 > 0.8$; at least five frames). Thus, $MSD_i$ ($R^2 > 0.8$) only uses around 4% of the data. Since the tiny fraction of the dataset that is used for analysis is chosen based on how well it fits an MSD-curve and since displacements of bound molecules are dominated by localization errors and therefore generally poorly fit by MSD-analysis, the procedure enriches for the free population, which is why the estimated bound fraction (16%) is so much lower than the true bound fraction (75%). Additionally, we note that $MSD_i$-based analysis is extremely sensitive to the fitting threshold: if instead of $R^2 > 0.8$, all trajectories had been used the estimated bound fraction would be 87% instead of 16%.

In conclusion, $MSD_i$ ($R^2 > 0.8$) is unreliable for estimating $F_{BOUND}$ when short trajectories are at stake, which is the usual case when performing intracellular SPT of fast-diffusing molecules. $MSD_i$ ($R^2 > 0.8$) most likely fails due to a combination of the following reasons among others. First, it poorly handles localization errors, which dominate the displacements of bound molecules. Second, by only considering trajectories of a certain length (normally at least five frames), it only analyzes a small subsample of the dataset. Third, there is no correction for defocalization bias. Since fast-diffusing molecules move out-of-focus and thus have shorter

trajectories, the 5-frame threshold introduces a large bias against freely-diffusing molecules. Fourth, the fitting threshold ($R^2>0.8$) is relatively arbitrary and the results of the analysis is extremely sensitive to this threshold. Accordingly, in these simulations $MSD_i$ ($R^2>0.8$) only analyzes a small fraction (~5%) of all the trajectories; note that this bias against the bound population provides a compensatory bias against the bound population to account for the bias against the free population due to defocalization bias. Fifth, it is difficult to justify the use of Gaussian distributions. Even in cases where the CDF-fit to the data is excellent, the fitted $F_{BOUND}$-value is often very far off the ground truth. Thus, the goodness of the fit cannot be used to judge how well the parameter-estimation went. Finally, we note that several variants of the $MSD_i$-based method exist (e.g. the approach used by Zhen *et al.* (*Zhen et al., 2016*)) is a bit different than the one used here. However, a full validation test of all $MSD_i$-based methods is beyond the scope of this work.

## $MSD_i$ (all)

### Rational and parameters

The $MSD_i$ (all) analysis was identical to $MSD_i$ ($R^2>0.8$) except for a single difference: instead of only using trajectories of at least five frames where the MSD-fit to individual trajectories was judged good ($R^2>0.8$), all trajectories of at least five frames were used, regardless of how good the MSD-fit was. five parameters were estimated in the fit.

### Performance evaluation

$MSD_i$ (all) analysis performed very poorly both when it comes to estimating $D_{FREE}$ and $F_{BOUND}$. The mean relative error (bias) for estimating $D_{FREE}$ was $-39.6\%$, the inter-quartile range (IQR) was 19.0% and the standard deviation 41.8%. For estimating $F_{BOUND}$ the bias for $MSD_i$ (all) was 22.0%, the IQR 17.8% and the standard deviation 15.8%. Thus, in all but a few edge cases, $MSD_i$ (all) cannot reliably estimate $D_{FREE}$ or $F_{BOUND}$. As for $MSD_i$ ($R^2>0.8$), examples of fits are shown in *Figure 3—figure supplement 3* and the Matlab code to perform the fitting is available together with the data (see "Data availability'). In the case of $MSD_i$ (all), the main reason for the unreliable estimates is due to defocalization bias. Since fast-diffusing molecules move out-of-focus and thus have shorter trajectories, the 5-frame threshold introduces a large bias against freely-diffusing molecules. Overall, consistent with previous benchmarking efforts on membrane proteins (*Weimann et al., 2013*), $MSD_i$ (all) performed least well among the tested methods.

## vbSPT

### Rational and parameters

vbSPT performs single-trajectory classification using Hidden-Markov Modeling (HMM) and Bayesian inference (*Persson et al., 2013*) and can assign different segments of a single trajectory to different diffusive states, each associated with a particular diffusion constant. vbSPT uses the information from all the estimates on single trajectories to consolidate an estimate of diffusion coefficients and associated fractions in each state.

vbSPT additionally uses a statistical model to infer the most likely number of diffusive states assuming all states to exhibit Brownian motion. Since the simulations used to evaluate vbSPT performed contain only two states, it was not clear how to assign $D_{FREE}$ or $F_{BOUND}$ in cases where e.g. three diffusive states were inferred. Therefore, to optimize the performance of vbSPT and perform the fairest comparison, we restricted vbSPT to two states such that vbSPT would infer the diffusion coefficient of up to two states and provide the associated fractions. This method conceptually differs from the $MSD_i$ approach in several ways:

- The inferred parameters are not based on the MSD
- A specific and rigorous Bayesian statistical model is used to aggregate the parameters estimated on single trajectories to global diffusion states.

vbSPT was initially designed for SPT of diffusing proteins in bacteria (*Persson et al., 2013*), where defocalization biases are virtually nonexistent since the axial dimension of most bacteria are generally comparable to or smaller than the microscope axial detection range. Furthermore, vbSPT does not explicitly model the localization error. It is then expected that the software performs poorly when the localization error is high, as can be expected when imaging intranuclear factors.

In practice, the following parameters were used to assess vbSPT performance. The software was run on the full set of 3480 simulations. The priors and optimization parameters were left as default and the scripts to perform the analysis are provided together with the experimental data (please see Data Availability section):

dim = 2;
trjLmin = 2;
runs = 3;
maxHidden = 2;
bootstrapNum = 10;
fullBootstrap = 0;
init_D = [0.001, 16];
init_tD = [2, 20]*timestep;

## Performance evaluation

Over the 3480 simulations, vbSPT accurately estimated both $D_{\text{FREE}}$ and $F_{\text{BOUND}}$. The mean relative error (bias) for estimating $D_{\text{FREE}}$ was 0.8%, the inter-quartile range (IQR) was 6.8% and the standard deviation 12.5%. For estimating $F_{\text{BOUND}}$ the bias for vbSPT was 5.0%, the IQR 6.1% and the standard deviation 4.6%. Thus, vbSPT estimated values were quite consistent (IQR <7% for both $D_{\text{FREE}}$ and $F_{\text{BOUND}}$). These values were very close to Spot-On in performance.

When looking at the heatmaps (*Figure 3—figure supplement 2*) more closely, it appeared that vbSPT performs poorly on the estimation of the free diffusion constant when the mean displacements are small. This case occurs either with small free diffusion constants (0.5–2 μm²/s), or with short frame rates (1 ms) and could be explained by the fact that in such conditions, the displacements of the free population and localization error have comparable magnitudes, and that vbSPT does not account for localization error.

Regarding the estimate of the fraction bound, vbSPT tends to overestimate it more and more as the mean displacement of the free population increases (that is, either the exposure time or $D_{\text{FREE}}$). This is most likely because vbSPT does not correct for defocalization bias. Thus, the more free molecules diffuse out-of-focus, the more vbSPT will overestimate $F_{\text{BOUND}}$. Finally, we note that these two biases somewhat compensate for each other: not considering localization errors causes a small overestimate of the free population, whereas not correcting for defocalization bias causes an underestimate of the free population.

In summary, for conditions where the mean jump length of the free population can be distinguished from the localization error, vbSPT performs reasonably well, while being slightly outperformed by Spot-On.

## Comparison of methods for 3-state simulations

In the case of the 720 simulated SPT datasets for the 3-state comparison, we analyzed the data using the Matlab version of Spot-On (all) and vbSPT. We describe the analysis in details below.

## Spot-On (all)

### Rational and parameters

Spot-On (all) was run on the simulations identically to the 2-state situation above except with one added freely diffusive state. To fit the simulations using Spot-On we fed the following parameters to the function SpotOn_core.m (v1.0; GitLab tag 1f9f782b):

- dZ = 0.700;
- GapsAllowed = 1;
- BinWidth = 0.010;
- UseAllTraj = 1;
- MaxJump = 6.05;
- ModelFit = 2;
- DoSingleCellFit = 0;
- NumberOfStates = 3;
- FitIterations = 8;
- FitLocError = 0;
- LocError = 0.04;
- D_Free1_3State = [0.4 10];
- D_Free2_3State = [0.4 25];
- D_Bound_3State = [0.00001 0.04];
- TimePoints: 10 if 1 ms; 9 if 4 ms; 8 if 7 ms; 7 if 10 ms; 6 if 12 ms; 5 if 20 ms;
- The empirical a,b parameters used to correct for defocalization bias were as follows:
  - $\Delta\tau$ = 1 ms; $\Delta z$ = 0.7 µm; 1 gap: a = 0.0387 $s^{1/2}$; b = 0.3189 µm;
  - $\Delta\tau$ = 4 ms; $\Delta z$ = 0.7 µm; 1 gap: a = 0.1472 $s^{1/2}$; b = 0.2111 µm;
  - $\Delta\tau$ = 7 ms; $\Delta z$ = 0.7 µm; 1 gap: a = 0.1999 $s^{1/2}$; b = 0.2058 µm;
  - $\Delta\tau$ = 10 ms; $\Delta z$ = 0.7 µm; 1 gap: a = 0.2379 $s^{1/2}$; b = 0.2017 µm;
  - $\Delta\tau$ = 13 ms; $\Delta z$ = 0.7 µm; 1 gap: a = 0.2656 $s^{1/2}$; b = 0.2118 µm;
  - $\Delta\tau$ = 20 ms; $\Delta z$ = 0.7 µm; 1 gap: a = 0.3133 $s^{1/2}$; b = 0.2391 µm;

As above, CDF-fitting was performed and the diffusion constant- and subpopulation fraction estimates compared to the ground truth for each of the 720 simulations for which the ground truth is known. Five parameters were estimated in the fit.

## Performance evaluation

As in the 2-state comparison, Spot-On (all) slightly, but significantly, outperformed vbSPT also in the case of 3 states. The biggest error (bias) in estimating any of the subpopulation fractions was 3% and the biggest standard deviation (3.6% std) was also small (see *Figure 3—figure supplement 6* for a full table for statistics). In the case of the diffusion constants, Spot-On also accurately inferred all of these with minimal error. The main limitation of Spot-On 3-state fitting, is that it sometimes gets stuck in local minima (we estimate this happens in <1% of cases). Therefore, it was necessary to increase the number of fitting iterations to 8. Nevertheless, Spot-On was very robust and accurately estimated all five parameters with minimal error and outperformed vbSPT.

## **vbSPT**

### Rational and parameters

vbSPT analysis was performed exactly as in the 2-state case, except with three hidden states instead of 2:

    dim = 2;
    trjLmin = 2;
    runs = 3;
    maxHidden = 3;
    bootstrapNum = 10;
    fullBootstrap = 0;
    init_D = [0.001, 16];
    init_tD = [2, 20]*timestep;

Although vbSPT was constrained to three states, it occasionally inferred that only 1 or 2 states exist. In case vbSPT inferred less than three states (1 or 2), the inferred diffusion coefficients were matched to the closest diffusion coefficient of the ground truth, and the proportion of the one or two unmatched diffusion coefficients was set to zero.

### Performance evaluation

vbSPT generally performed quite well. The maximal error (bias) in estimating any of the subpopulation fractions was 6% and the maximal standard deviation (6.3% std; see *Figure 3— figure supplement 6* for a full table for statistics). The main limitation of vbSPT was its inability to infer $D_{SLOW}$: the mean error (bias) for estimating $D_{SLOW}$ was 36.6% and the standard deviation was 64.7%. Therefore, vbSPT performed almost as well as Spot-On for estimating the subpopulation fractions and for estimating $D_{FAST}$, but vbSPT was unable to accurately estimate both $D_{BOUND}$ and $D_{SLOW}$ and thus failed when estimating 2 out of the five parameters. In conclusion, vbSPT performs almost as well as Spot-On when estimating subpopulation fractions, but quite poorly when estimating diffusion constants unless they are very high.

## Appendix 2

DOI: https://doi.org/10.7554/eLife.33125.028

# Considerations for choosing Spot-On parameters

In order to run Spot-On, the user has to set a number of parameters. While some are determined by the acquisition protocol (e.g. time between frames), others will have to be carefully chosen. We provide a discussion of how to choose these here.

## JumpsToConsider

Users can either choose to use all displacements from all trajectories (set 'Use all trajectories' to 'Yes' in the web-version of Spot-On or 'UseAllTraj = 1' in the Matlab version of Spot-On) or to use only a subset by controlling the JumpsToConsider variable. For example, consider a trajectory consisting of 6 localizations and without gaps. If JumpsToConsider = 4 and TimePoints = 6, then this trajectory will contribute four displacements to the $1\Delta\tau$ histogram, four displacements to the $2\Delta\tau$ histogram, three displacements to the $3\Delta\tau$ histogram, two displacements to the $4\Delta\tau$ histogram and one displacement to the $5\Delta\tau$ histogram. Thus, even though the trajectory contains 5 $1\Delta\tau$ displacements, only the first four will be used for analysis if JumpsToConsider = 4. Why would we want to limit the number of jumps that were used? Since freely-diffusing molecules move out-of-focus, almost all very long trajectories will be bound molecules. For example, a single trajectory of 21 localizations will provide 20 displacements to the $1\Delta\tau$ histogram, whereas freely diffusing molecules with short trajectories will provide fewer (e.g. 10 trajectories with three localizations would be necessary to also provide 20 displacements to the $1\Delta\tau$ histogram). Thus, by limiting JumpsToConsider, one is biasing the displacement histogram against bound molecules. However, as demonstrated in the simulations shown in *Figure 3—figure supplement 2*, whether all jumps or JumpsToConsider = 4 is used has almost no effect on the $D_{\text{FREE}}$-estimate, but using JumpsToConsider = 4 causes $F_{\text{BOUND}}$ to be underestimated by on average of $-5\%$ (percentage points) relative to SpotOn (all). We see a similar ~5–10% difference between Spot-On (four jumps) and Spot-On (all) on the experimental spaSPT data shown in *Figure 4*. As we have discussed previously (*Hansen et al., 2017*), restricting JumpsToConsider to four is a way one can compensate for all the many acquisition biases (such as motion-blur) that generally cause undercounting for fast-diffusing molecules and which cannot readily be taken into account in simulations. While the optimal value will depend on the trajectory length distribution (JumpsToConsider should not take a value much smaller than the mean trajectory length), we found that JumpsToConsider = 4 provides a good compromise for our experimental data. We strongly recommend including experimental controls (such as histone H2B-Halo and Halo-3xNLS to ensure that experimental and analysis parameters have been reasonably set).

## Number of timepoints

Spot-On considers how the histogram of displacement changes over time for multiple $\Delta\tau$. The number of $\Delta\tau$ that will be considered is equal to the number of timepoints – 1. So, if timepoints = 8, the displacements from $1\Delta\tau$ to $7\Delta\tau$ will be considered. How many timepoints to consider will depend on how much data you have and the frame-rate. For example, if the mean trajectory length is two frames, setting timepoints to 20 will cause problems since only a tiny fraction of trajectories will be at least 20 frames long and thus contribute to the $19\Delta\tau$ histogram. Moreover, the correction for defocalization is approximate, so considering timepoints where more than >95% of free molecules have moved out-of-focus is also not recommended; when this happens will further depend on the free diffusion constant. Nevertheless, as long as there is sufficient data to reasonably populate the displacement histograms at all timepoints, Spot-On is highly robust to how this parameter is set (*Figure 3—*

*figure supplement 8*). As a rule of thumb we generally do not recommend setting timepoints above 10 or considering $\Delta\tau$ beyond 80 ms.

### Iterations for fitting

Spot-On almost always converges optimally in the first iteration, so generally 2 or three is more than sufficient when using the 2-state model. For the 3-state model, the parameter estimation is more complicated and here we recommend eight iterations as a starting point.

### PDF or CDF fitting

Although for large datasets PDF- and CDF-fitting perform similarly as shown in *Figure 3—figure supplement 9*, CDF-fitting tends to provide more reliable estimates of $D_{\text{FREE}}$ and $F_{\text{BOUND}}$ when the number of trajectories decreases, likely because PDF-fitting is more susceptible to binning noise. Thus, for quantitative analysis we always recommend CDF-fitting, though PDF-fitting can be convenient for making figures since most people find histograms more intuitive.

### Fitting localization error

Spot-On can either use a user-supplied localization error or fit it from the data. As long as there is a significant bound fraction, Spot-On will infer this with nanometer precision (*Figure 3—figure supplement 11*), though we note that this is an average localization error that mostly reflects the localization error of the bound fraction, and the actual localization error for each individual localization will vary (*Deschout et al., 2012*; *Lindén et al., 2017*). In cases, where the bound population is very small, fitting the localization error can be less accurate. Thus, in situations where comparisons are being made between the same protein under different conditions or e.g. between different mutants of the same protein, we recommend fitting to obtain a mean localization error and then keeping it fixed in the comparisons.

### Choosing allowed ranges for diffusion constants

Spot-On comes with default allowed ranges. For example, for the 2-state model, $D_{\text{FREE}} = [0.5; 25]$ and $D_{\text{BOUND}} = [0.0001; 0.08]$. These ranges are generally reasonable, but may not be appropriate for all datasets. Whenever Spot-On infers a diffusion constant that is equal to the min or max, caution is needed and it may be necessary to change these limits. In particular, unless a molecule is bound to an unusually dynamic scaffold, $D_{\text{BOUND}}$=0.08 μm$^2$/s is almost certainly too high. Thus, we recommend imaging a protein that is overwhelmingly bound, such as histone H2B or H3, fitting the histone data with Spot-On and then use the inferred $D_{\text{BOUND}}$ for histone proteins or a slightly larger value as the maximally allowed $D_{\text{BOUND}}$ value.

### 2-state or 3-state model

Spot-On considers either a 2-state or 3-state model. Since the 3-state model contains two additional fitted parameters, the 3-state fit is almost always better. While there are many cases where a 2-state model would be inappropriate (e.g. a transcription factor that can exist as either a monomer or tetramer, thus exhibiting two very different diffusive states), generally speaking, we prefer fitting a 2-state model for most transcription factors or similar nuclear chromatin-interacting proteins. In part, deviations from the 2-state model will be due to anomalous diffusion and confinement inside cells, which cause deviation from the ideal Brownian motion model implemented by Spot-On. For this reason, traditional model-selection techniques such as Akaike's Information Criterion (AIC) or the Bayesian Information Criterion (BIC) can also be misleading.

## Appendix 3

DOI: https://doi.org/10.7554/eLife.33125.029

# SPT acquisition considerations in spaSPT experiments

## Considerations for minimizing bias in SPT acquisitions

To obtain a good single-molecule tracking dataset, a series of requirements have to be met. First of all, it must be possible to image single-molecules at a high signal-to-noise ratio. This is now relatively straightforward thanks to developments in fluorescence labeling strategies and imaging modalities (*Lavis, 2017*; *Liu et al., 2015*). The development of the HaloTag protein-labeling system and bright, photo-stable organic Halo-dyes such as TMR and the JF dyes (*Grimm et al., 2015*) now make it possible to easily visualize single protein molecules inside live cells. Moreover, imaging modalities such as highly inclined and laminated optical sheet illumination ('HiLo')(*Tokunaga et al., 2008*) are relatively straightforward to implement and combined with a high-quality EM-CCD camera make it possible to image single-molecules at high signal-to-noise suitable for generating high-quality 2D SPT data. For details of our imaging setup, which combines HaloTag-labeling with HiLo-illumination and which is relatively common and easy to operate, please see the methods section. But we note that many other imaging modalities, e.g. light-sheet or even epi-fluorescence imaging can generate high-quality single-molecule tracking data.

Thus, in the following we will assume that the above condition is met: namely, that single protein molecules can be tracked inside live cells at high signal-to-noise ratio. Nevertheless, even if this condition is met, there are at least four other major sources of bias:

1. Detection: minimize 'motion-blurring'
2. Tracking: minimize tracking errors
3. 3D loss: correct for molecules moving out-of-focus (defocalization bias)
4. Analysis methods: infer subpopulations with minimal bias

Spot-On addresses point 3 and 4, as described elsewhere, but point 1 and 2 must be addressed in the experimental design. We discuss strategies to minimize these biases below (spaSPT).

## 1. Detection – minimizing 'motion-blurring'

Almost all localization algorithms achieve sub-diffraction localization accuracy ('super-resolution') by treating individual fluorophores as point-source emitters, which generate blurred images that can be described by the Point-Spread-Function (PSF) of the microscope. Modeling of the PSF (typically as a 2-dimensional Gaussian) then allows extraction of the particle centroid with a precision of tens of nanometers. But as illustrated in *Figure 1A*, while this works extremely well for bound molecules, fast-diffusing molecules will spread out their photons over many pixels during the camera exposure and thus appear as 'motion-blurs'. Thus, localization algorithms will reliably detect bound molecules, but may fail to detect fast-moving molecules as has also been observed previously (*Berglund, 2010*; *Deschout et al., 2012*; *Elf et al., 2007*; *Izeddin et al., 2014*; *Lindén et al., 2017*). Clearly, the extent of the bias will depend on the exposure time and the diffusion constant: the longer the exposure and higher $D$, the worse the problem. Assuming Brownian motion, we can calculate the fraction of molecules that will move more than some distance, $r_{\max}$, during an exposure time, $t_{\exp}$, given a free diffusion constant of $D_{\mathrm{FREE}}$ using the following equation:

$$P(r > r_{\max}) = e^{\frac{-r_{\max}^2}{4D_{\mathrm{FREE}}t_{\exp}}}$$

For example, if we define motion-blurring as moving more than two pixels (>320 nm assuming a 160 nm pixel size) during the excitation, an exposure time of 10 ms and a typical free diffusion constant of 3.5 μm$^2$/s (e.g. ~Sox2), we get:

$$P(r>0.32\mu m) = e^{\frac{-(0.32\mu m)^2}{4\cdot 3.5\frac{\mu m^2}{s}\cdot 0.010s}} = 0.48$$

Thus, even for a relatively slowly diffusing protein, with a 10 ms exposure we should expect almost half (48%) of all free molecules to show significant motion-blurring, if we assume that molecules move with a constant speed during the exposure. The most straightforward solution, therefore, is to limit the exposure time: in the limit of an infinitely short exposure time, there is no motion-blur. In practice, most EM-CCD cameras can only image at ~100–200 Hz for reasonably sized ROIs. Moreover, it is generally desirable for the mean jump lengths to be significantly bigger than the localization error, thus for most nuclear factors in mammalian cells it is not desirable to image at above >250 Hz. Accordingly, a reasonable solution is therefore to use stroboscopic illumination. That is, using brief excitation laser pulses that last shorter than the camera frame rate (e.g. 1 ms excitation pulse, 10 ms camera exposure time for a 100 Hz experiment): this achieves minimal motion-blurring while maintaining a useful frame-rate. However, this highlights a key experimental trade-off: shorter excitation pulses minimize motion-blurring, but also minimize the signal-to-noise. Therefore, a reasonable compromise has to be determined. Here we use 1 ms excitation pulses: this achieves minimal motion blurring (0.067% > 320 nm using $D$ = 3.5 µm$^2$/s) and still yields very good signal (signal-to-background >5). But users will need to decide this based on their expected $D$ and their experimental setup (signal-to-noise). Moreover, different localization algorithms (*Chenouard et al., 2014*; *Deschout et al., 2012*) have different sensitivities to motion-blurring; thus, the extent of the bias will also depend on the user's localization algorithm. As we show here, in the case of the MTT-algorithm (*Sergé et al., 2008*), the estimation of $D$ is quite sensitive to motion-blurring, but the estimation of the bound fraction is less sensitive as long as the diffusion constant is <5 µm$^2$/s. But other localization algorithms may be more or less sensitive. Generally speaking, we do not recommend imaging at a signal-to-background <3 and do not recommend using excitation pulses >5 ms, but the optimal conditions will need to be determined on a case-by-case basis.

In conclusion, experimentally implementing stroboscopic excitation makes it possible to minimize the bias coming from motion-blurring, while still achieving a sufficient signal for reliable localization.

## 2. Tracking – minimizing tracking errors

It is necessary to minimize tracking errors in order to obtain high-quality SPT data. Tracking errors bias the estimation of essentially all parameters we could want to estimate from SPT experiments including diffusion constants, subpopulations, anomalous diffusion etc. While many different tracking algorithms exist, it is fundamentally impossible to perform tracking, that is connecting localized molecules between subsequent frames, at high densities without introducing many tracking errors. Thus, the simplest solution is to image at low densities: in principle, if there is only one labeled molecule per cell, there can be no tracking errors. Yet, because dyes generally bleach quite quickly under most SPT imaging conditions, this has traditionally led to a serious trade-off between data quality and the number of trajectories which can be obtained. However, with the recent development of bright photo-activatable JF-dyes (*Grimm et al., 2016a*; *2016b*) (PA-dye), it is now possible to combine the superior brightness of the Halo-JF dyes with photo-activation SPT (also called sptPALM (*Manley et al., 2008*)). That is, a large fraction of Halo-tagged proteins in a cell can be labeled with Halo-PA-JF dyes and then photo-activated one at a time: this allows imaging at extremely low densities (<1 fluorescent molecule per cell per frame) and nevertheless tens of thousands of trajectories from a single cell can be obtained. Thus, PA-dyes now make it possible to nearly eliminate tracking errors without compromising on signal-to-noise or amount of data. In fact, imaging at extremely low densities generally also improves signal-to-noise since out-of-focus background is reduced and overlapping point emitters are avoided (*Izeddin et al., 2014*).

Nevertheless, even with paSPT it is still necessary to decide on an optimal density. The key parameters are size of the ROI (ideally the whole nucleus for studies in cells) and $D$: a large

nucleus and a slow *D* can support a higher density than fast-diffusing molecules in a small nucleus. As a general rule of thumb, we recommend a density of ~1 fluorescent molecule per ROI per frame. This will keep tracking errors at a minimum and still support rapid acquisition of large datasets. All data acquired for this study was acquired at approximately this density.

In practice, keeping an optimal density will require some trial-and-error optimization of the 405 nm photo-activation laser intensity. 405 nm excitation does contribute background fluorescence, so we prefer to pulse the 405 nm laser during the camera 'dead-time' (~0.5 ms in our case) to avoid this. Moreover, this also makes it easier to keep the photo-activation level constant when changing the frame rate. However, the optimal photo-activation power will depend on the expression level of the protein, protein half-life and the dye concentration and will therefore have to be optimized in each case. We recommend recording initial datasets and then analyzing them using Spot-On which reports the mean number of localizations per frame and then using this information to determine the optimal photo-activation level. However, even then some cell-to-cell variation may be unavoidable: especially in transient transfection experiments where there is large cell-to-cell variation in expression level or when studying proteins expressed from stably integrated transgenes (e.g. Halo-3xNLS and H2b-Halo in our case). In these cases, some cells will likely exhibit too high a density. To deal with this, Spot-On includes the option to analyze datasets from individual cells first and then excluding a cell with too high a density before analyzing the merged dataset.

## Which datasets are appropriate for Spot-On?

In the sections above, we have discussed how to minimize common experimental biases in SPT experiments and proposed spaSPT as a general solution. However, many 2D SPT datasets recorded under different conditions are also appropriate for Spot-On. For example, SPT experiments without photo-activation or with continuous illumination may also be appropriate for analysis with Spot-On. For example, there may be situations where photo-activation SPT is not possible: in such cases, it will be essential to keep the labeling density sufficiently low that tracking errors are minimized and it might thus be necessary to image substantially more cells to get enough statistics. Likewise, as we show in *Figure 4JK*, motion-blurring is a major concern for fast-diffusing molecules, but for a slowly diffusing molecule like Halo-CTCF it makes only a small difference. Thus SPT datasets recorded with continuous illumination may also be appropriate provided that the protein of interest is known to diffuse sufficiently slowly.

We also note that since Spot-On uses the loss of fast-diffusing molecules over time to correct for bias and to estimate the free population, it is essential that all trajectories are included in Spot-On for analysis. For example, some tracking and localization algorithms ignore all trajectories below a certain length (e.g. five frames), but this will cause Spot-On to misestimate the loss of molecules moving out-of-focus and thus it is imperative that trajectories of all lengths be included when analyzing data using Spot-On. Furthermore, trajectories of only a single localization are required to accurately compute the average number of localizations per frame, which is a key quality-control metric for SPT data.

Moreover, Spot-On does not currently support 3D SPT data. Furthermore, Spot-On assumes diffusion to be Brownian. This is a reasonable approximation even for molecules exhibiting some levels of anomalous diffusion as shown in *Figure 4—figure supplement 2*, but Spot-On is not appropriate for molecules undergoing directed motion (e.g. a protein moving on microtubules). Additionally, in cases where there are frequent state transitions at a time-scale similar to the frame rate (e.g. transcription factor with a 10 ms residence time imaged at 100 Hz), Spot-On may give inaccurate results since it ignores state transitions (*Figure 3—figure supplement 10*). Finally, the correction for molecules moving out-of-focus assumes that molecules are not fully confined within small compartments, that prevent molecules from moving out-of-focus.

## Appendix 4

DOI: https://doi.org/10.7554/eLife.33125.030

# Proposed minimal reporting guidelines for SPT data and kinetic modeling analysis

To ensure reproducibility of results and subsequent analyses, datasets, statistics and analysis metrics should be provided. This should allow the reader to quickly assess the quality and statistical significance of the presented results and datasets. So far, to our knowledge, no consensus exists on minimal reporting guidelines for single particle tracking datasets and kinetic modeling analyses. We note, however, that a recent preprint suggests a similar conceptual framework, although less applicable to single-molecule experiments (*Rigano and Strambio De Castillia, 2017*),

We propose that published single-particle datasets be published and reported accompanied with the following metadata. We suggest that these metrics constitute a minimal reporting guideline for single-particle datasets and subsequent kinetic modeling (though additional information may be appropriate and necessary in some cases).

## Dataset description

| Criterion | How to obtain it | Example value |
|---|---|---|
| Exposure time | Determined at the acquisition step | 5 ms |
| Signal-to-background ratio | Mean peak value of detected particle divided by mean background value | 5 |
| Detection algorithm used | | MTT (version xxx) |
| Tracking algorithm used | | MTT (version xxx) |
| Number of particles per frame | Provided by Spot-On | Mean: 0.76 |
| Number of detections | Provided by Spot-On | 360000 |
| Number of trajectories of length >3 | Provided by Spot-On | 10000 |
| Mean trajectory length | Provided by Spot-On | 4.5 frames |
| Localization error | Provided by Spot-On | 30 nm |

## Spot-On parameters

In addition to these metrics, it is important to report the parameters specified in the detection and tracking algorithms, since this can greatly affect the results. For Spot-On, we recommend reporting the following parameters:

- Jump length distribution parameters: BinWidth (μm), Number of timepoints, Jumps to consider or Use all trajectories, MaxJump (μm),
- Fitting parameters: Number of states (2 or 3), localization error fitted from data (Yes or No, if no, specify the value, in nm), dZ (μm), a ($s^{-1/2}$), b (μm), PDF or CDF fit (PDF or CDF), number of iterations. Finally, the bounds used for the fitting algorithm should be reported, e.g:
  - $D_{bound}$: [0.0005, 0.08] $μm^2/s$
  - $D_{free}$ [0.15, 25] $μm^2/s$
  - $F_{bound}$ [0,1]
  - Obviously, if a 3-state model is used, the bounds for the additional subpopulation should also be reported.

In case a custom-modified version of Spot-On is used, we recommend that the code be made available and that a summary of the modifications be included in the methods section.

