## [Decision Letter]

Thank you for submitting your article "Spot-On: robust model-based analysis of single-particle tracking experiments" for consideration by *eLife*. Your article has been reviewed by two peer reviewers, and the evaluation has been overseen by a Reviewing Editor and Kevin Struhl as the Senior Editor. The following individual involved in review of your submission has agreed to reveal his identity: Lothar Schermelleh (Reviewer #2).

The reviewers have discussed the reviews with one another and the Reviewing Editor has drafted this decision to help you prepare a revised submission.

This 'Resource' manuscript describes an integrated approach for single-particle tracking (SPT) microscopy using stroboscopic photoactivation (spaSPT) and present "Spot-On", an open-access software analysis tool for SPT data. The latter can be applied to any kind of SPT data and uses model fitting to calculate diffusion rates and bound and mobile population sizes. The authors convincingly validate and compare their approach with other available tools using ground-truth simulations and analysing a number of Halo-tagged nuclear proteins with differential dynamic properties in U2OS cells and labelled with photoactivatable JF dyes.

Both reviewers are enthusiastically supportive of this being published, although a small number of minor comments and corrections are appended below.

1) A practical issue – could some of the information on the website also be loaded as a supplemental tutorial file (some helpful screenshots?)?

2) The software in its current form is restricted to 2D-SPT analyses. Why is 3D-tracking not implemented, and is there a planned route for a future upgrade? With many imaging systems offering astigmatism-based 3D localisation option, would this not offer potential benefits? We understand that a package for 3D analysis may be out of the scope of the present article, but whether there is a prospect for such a package in the future might be indicated/commented on.

3) Along with the software, the authors describe spaSPT as a beneficial approach to minimize motion-blur and tracking errors. What would be the difference between spaSPT and SPT with very low concentration of non-photoactivatable JF dyes? The authors may want to discuss application of Spot-On with alternative imaging approaches in cases where photoactivation is either not possible (e.g. due to lack of laser lines or suitable dyes) or not desired (potential damage, blue channel is already used).

4) It would be desirable to have a function for visualising tracks with the option to differentially select and display subsets of tracks that fall into different categories (immobile/mobile or other criteria). This would enable the user to directly see spatial patterns of differential dynamics.

5) i) Figure 4 and subsection “Effect of motion-blur bias on parameter estimates”, first paragraph: Why is there is a small but significant difference between PA-JF_549_ and PA-JF_646_?

ii) Subsection “Validation of Spot-On using spaSPT data at different frame rates”: The axial detection range should be specified (around 700 nm?).

iii) Abbreviations CDF and PDF may need to be introduced in the main text as well, not only the figure legend.

iv) Figure 4—figure – supplement 3 legend – last paragraph: Does the "total bound fraction of ~60-65%" refer to Halo-*Sox2*? Please specify.

---

## [Author Response]

[…] Both reviewers are enthusiastically supportive of this being published, although a small number of minor comments and corrections are appended below.1) A practical issue – could some of the information on the website also be loaded as a supplemental tutorial file (some helpful screenshots?)?

This is a good suggestion. We adapted the online tutorial and attach it as Supplementary file 1. To accommodate future revisions of the interface, it includes a reference to the online version, that will follow the upgrades of the platform.

2) The software in its current form is restricted to 2D-SPT analyses. Why is 3D-tracking not implemented, and is there a planned route for a future upgrade? With many imaging systems offering astigmatism-based 3D localisation option, would this not offer potential benefits? We understand that a package for 3D analysis may be out of the scope of the present article, but whether there is a prospect for such a package in the future might be indicated/commented on.

This is a very interesting suggestion and one we have thought a lot about. Currently, there are two main options for 3D tracking: 1) PSF shaping (such as astigmatism)-based 3D SPT (e.g. cylindrical lens or adaptive optics) and 2) 3D SPT using a Multi-Focal Microscope (MFM) (Abrahamsson et al., Nature Methods, 2013). We will discuss both here.

1) astigmatism-based 3D SPT data will give a truncated 3D displacement distribution. In x,y there are minor restrictions on displacement lengths, but in z, the max displacement will be equal to the axial detection range (~700 nm total; 0 +/- 350 nm). For example, a molecule 200 nm above the focal plane, will defocalize if it moves more than 150 nm up or more than 550 nm down. For this reason, the expected distribution of 3D displacements is not the simple 3D Brownian distribution, but a complicated convolution of a truncated axial distribution (max 700 nm; position-dependent) and the 2D x,y-distribution considered by Spot-On. To our knowledge there is no straightforward way of extending the z-correction currently implemented in Spot-On to 3D. The simulations in Author response image 1 compare the jump length distributions derived from data simulated (D=6.0µm²/s, dt=13ms, sigma=35nm) inside a nucleus of diameter 8µm with increasing axial detection range (400nm, 700nm and full nucleus). Although (right panel) the 2D projection from simulated data is only marginally affected by changes in the axial detection depth, in 3D (left panel), the jump length distribution is sensitive to the axial detection depth, and the lower the axial detection depth, the more truncated the distribution. More work is necessary to determine how mathematically tractable this problem is (and we agree that this will be an interesting future direction), but currently we are not sure if the additional information justifies the approach and of course 3D-astigmatism data can always be 2D-projected and then analyzed by Spot-On.

2) the truncated 3D displacement distribution issue can in theory be solved with MFM which can in principle yield whole-nucleus 3D SPT data. However, this comes with serious limitations. First, in the 9-focal plane implementation, the signal is split into 9 and due to other losses, the signal-to-noise is reduced by about a factor of ~15-20. Although the JF-dyes are very bright, this big a loss of signal is a serious limitation. Second, since the full field-of-view is required, the frame rate is necessarily slow (33 Hz in Chen Cell 2014 with EM-CCD). While sCMOS cameras might solve the speed issue, this comes at the cost of further loss of signal. Thus, in MFM-based 3D SPT, continuous illumination was necessary to collect enough signal resulting in 30 ms continuous exposure time. This leads to serious motion-blurring and likely undercounting of the fast-moving population. Third, while incredibly elegant, the MFM microscope is quite complicated to set up (we have set it up in our lab recently) and not in regular use in many labs. But as the reviewers also suggest, this is likely to improve in the future, though we note that no Z-correction for defocalization would be necessary in whole-nucleus MFM.

In summary, we believe that 3D tracking methods are currently not sufficiently mature to provide enough added value. But we nevertheless fully agree with the reviewers that this is very likely to change in the future and we have now added a sentence to the Discussion that extending Spot-On to 3D SPT data is an exciting future direction and one that we are potentially interested in pursuing. Specifically, we now write that: “This platform can easily be extended to other diffusion regimes (Metzler et al., 2014) and models (Lee et al., 2017) and, as 3D SPT methods mature, to 3D SPT data.”

3) Along with the software, the authors describe spaSPT as a beneficial approach to minimize motion-blur and tracking errors. What would be the difference between spaSPT and SPT with very low concentration of non-photoactivatable JF dyes? The authors may want to discuss application of Spot-On with alternative imaging approaches in cases where photoactivation is either not possible (e.g. due to lack of laser lines or suitable dyes) or not desired (potential damage, blue channel is already used).

This is an important point and as the reviewers suggest, there is in principle no difference between doing SPT with very low concentrations of non-PA dye and spaSPT. It’s just that with non-PA dyes, once the dyes have bleached no more data can be obtained from the cell in question and thus, many more cells have to be imaged to obtain enough data. So spaSPT is more convenient. But to clarify this important point, we now write: “We also note that although Spot-On was validated on spaSPT data, SPT data with non-photoactivatable dyes is also suitable for Spot-On analysis provided that the density is sufficiently low to minimize tracking errors. (see also Appendix 3: “Which datasets are appropriate for Spot-On?”)”.

4) It would be desirable to have a function for visualising tracks with the option to differentially select and display subsets of tracks that fall into different categories (immobile/mobile or other criteria). This would enable the user to directly see spatial patterns of differential dynamics.

We thank the reviewers for this important consideration. Indeed, track visualization is crucial as it can be used both to (1) perform quality controls and spot several kinds of biases and (2) refine data analysis, for instance using track segmentation and we discuss both below.

Quality controls (1): visual inspection of the output of the tracking algorithm is indeed crucial, and can be used to detect biases such as tracking errors or issues with a detection threshold. In that case, such visual inspection is much more useful when trajectories and raw images can be overlayed. To draw the full potential of such approach, one would need to upload the raw data to Spot-On, in addition to the tracked data.

Although tracked datasets (trajectories) are easily amenable to online processing (a typical SPT tracking file has a size of the order of a few MB), raw images often weigh several GB, making their interactive processing more challenging. For this reason, we do not believe such an option will work for a web-based platform like Spot-On.

Conversely, several imageJ/Fiji plugins are capable of interactive display of raw images and trajectories. In our opinion, one of the most mature Fiji plugins is TrackMate (Tinevez et al., Methods, 2017). TrackMate can save/reopen tracking files that contain a full description of the tracking parameters, allowing for a careful inspection of the raw movies and overlaid trajectories.

In order to facilitate the integration between a track-visualization software (TrackMate) and a SPT analysis tool (Spot-On), we developed a “TrackMate to Spot-On connector” (available at: https://gitlab.com/tjian-darzacq-lab/Spot-On-TrackMate). This plugin adds an extra menu to TrackMate that allows a one-click upload of a dataset to Spot-On. Thus, a manually inspected file can automatically be uploaded to Spot-On.

We now mention this tool: “Spot-On does not directly analyze raw microscopy images, since a large number of localization and tracking algorithms exist that convert microscopy images into single-molecule trajectories (for a comparison of particle tracking methods, see (Chenouard et al., 2014); moreover, Spot-On can be interfaced with TrackMate (Tinevez et al., 2017), which allows inspection of trajectories before uploading to Spot-On).” Finally, we acknowledge that this approach restricts the inspection of trajectories to the ones tracked using the TrackMate software, and that a tool accepting a wider range of file formats would be desirable. Our group is currently pursuing efforts in that direction, but this is a longer-term project that is not within the scope of this paper.

Track segmentation/classification (2): even though Spot-On can infer diffusion constants and relative fraction of several subpopulations, it does not perform single-trajectory classification. Indeed, this latter problem is significantly harder, or even impossible, depending on the length of the track to classify (see for instance Michalet and Berglund, 2012), and several recently developed approaches already deal with this problem (Persson et al. 2013, Monnier et al., 2015), mostly relying on hidden Markov models (HMMs). Furthermore, since trajectories derived from 3D-diffusing factors entering and exiting the focal plane are inherently short, only a very small fraction of them are actually amenable to single-trajectory inference. Therefore, while we agree that assigning trajectories into particular subcategories (e.g. immobile vs. mobile) is interesting, since HMM-based approaches already exist, we feel that adding this functionality is outside the scope of this paper.

In summary, we thank the reviewers for the suggestion and have implemented a one-click open-source connector between TrackMate and Spot-On that makes visualizing the trajectories before uploading them to Spot-On easy and intuitive.

5) i) Figure 4 and subsection “Effect of motion-blur bias on parameter estimates”, first paragraph: Why is there is a small but significant difference between PA-JF_549_ and PA-JF_646_?

This is a good point and we have noticed this as well. For proteins with a significant bound fraction (i.e. >25%), we find that both PA-JF_549_ and PA-JF_646_ give identical results. However, we find that for proteins with a negligible bound fraction (e.g. Halo-3xNLS or Halo-CTCF without the DNA binding domain), PA-JF_646_ has a “bound fraction floor” of about 10-15% which it cannot go below, whereas PA-JF_549_ appears to be able to capture the dynamics of all proteins and we have observed bound fractions <5% for PA-JF_549_. We believe this is the reason for the difference and we are pretty confident in this effect: it has been highly reproducible in both U2OS and mES cells and we have observed it for several different proteins. We do not have a clear explanation for this, however; one possibility is that JF_646_ and its conjugates are a little more hydrophobic due to the extra methyl groups in the Si-rhodamine structure. This effect is unlikely to be a case of tracking “free dye ligand”, since JF_646_ shows negligible photoactivation unless bound to the HaloTag (see Grimm 2016) and since small dyes like JF_549/646_ are expected to have diffusion constants ~250 um2/s and thus are unlikely to be trackable. Since we do not have a clear explanation of this effect, we prefer not to speculate in the manuscript, but we now state that JF_549_ appears to be more reliable for SPT (which could be helpful to know for other labs) and specifically write that “Similar results were obtained for both dyes for proteins with a significant bound fraction, but we note that JF_549_ appears to better capture the dynamics of proteins with a minimal bound fraction such as Halo-3xNLS (Figure 4)”.

ii) Subsection “Validation of Spot-On using spaSPT data at different frame rates”: The axial detection range should be specified (around 700 nm?).

Yes, we agree and have updated the sentence. Thanks for pointing this out.

iii) Abbreviations CDF and PDF may need to be introduced in the main text as well, not only the figure legend.

Yes, this is another good point. We now added this to the Discussion around Figure 3 and state that: “Note that although we show the fits to the probability density function since this is more intuitive (PDF; histogram), we performed the fitting to the cumulative distribution function (CDF).”

iv) Figure 4—figure supplement 3 legend – last paragraph: Does the "total bound fraction of ~60-65%" refer to Halo-Sox2? Please specify.

Thanks for catching this error. The reviewer is correct; it referred to Halo-*Sox2* and we have now corrected this.